# ATA: Adaptive Task Allocation for Efficient Resource Management in Distributed Machine Learning

**Artavazd Maranjyan** [1]  **El Mehdi Saad** [1]  **Peter Richtárik** [1]  **Francesco Orabona** [1]

## Abstract

Asynchronous methods are fundamental for parallelizing computations in distributed machine learning. They aim to accelerate training by fully utilizing all available resources. However, their greedy approach can lead to inefficiencies using more computation than required, especially when computation times vary across devices. If the computation times were known in advance, training could be fast and resource-efficient by assigning more tasks to faster workers. The challenge lies in achieving this optimal allocation without prior knowledge of the computation time distributions. In this paper, we propose ATA (Adaptive Task Allocation), a method that adapts to heterogeneous and random distributions of worker computation times. Through rigorous theoretical analysis, we show that ATA identifies the optimal task allocation and performs comparably to methods with prior knowledge of computation times. Experimental results further demonstrate that ATA is resource-efficient, significantly reducing costs compared to the greedy approach, which can be arbitrarily expensive depending on the number of workers.

## 1. Introduction

In this work, we address a very general yet fundamental and important problem arising in various contexts and fields. In particular, there are $n$ workers/nodes/devices collaborating to run some iterative algorithm which has the following structure:

[1]King Abdullah University of Science and Technology (KAUST), Thuwal, Saudi Arabia. Correspondence to: Artavazd Maranjyan <arto.maranjyan@gmail.com>, El Mehdi Saad <mehdi.saad@kaust.edu.sa>, Peter Richtárik <peter.richtarik@kaust.edu.sa>, Francesco Orabona <francesco@orabona.com>.

*Proceedings of the $42^{nd}$ International Conference on Machine Learning*, Vancouver, Canada. PMLR 267, 2025. Copyright 2025 by the author(s).

- In order to perform a single iteration of the algorithm, a certain number ($B$) of *tasks* needs to be performed.

- Each task can be computed by any worker, and the tasks are not temporally related. That is, they can be computed in any order, in parallel, and so on.

- Whenever a worker is asked to perform a single task, the task will take a certain amount of time, modeled as a nonnegative random variable drawn from an unknown distribution specific to that worker. The stochastic assumption makes sense because in real systems computation times are not fixed and can vary with each iteration (Dean & Barroso, 2013; Chen et al., 2016a; Dutta et al., 2018; Maranjyan et al., 2025a).

- Each worker can only work on a single task at a time. That is, a worker processes all tasks it has to perform sequentially. Different workers work in parallel.

A natural goal in this setup is to make sure all tasks are completed as fast as possible (in expectation), which minimizes the (expected) time it takes for a single iteration of the algorithm to be performed provided that the task completion time is the dominant time factor of the iteration. Provided we are willing to waste resources, there is a simple solution to this problem, a Greedy Task Allocation (GTA) strategy, which follows this principle: *Make sure all workers are always busy working on some task, and stop once $B$ tasks have been completed.* In GTA, we initially ask all $n$ workers to start working on a task, and as soon as some worker is done with a task, we ask it to start completing another task. This process is repeated until $B$ tasks have been completed.

While GTA minimizes the completion time, it can be immensely wasteful in terms of the total worker utilization time needed to collect all $B$ tasks. Indeed, consider the scenario with $n = 1000$ workers and $B = 10$ tasks. In this case, GTA will lead to at least $n - B = 990$ unnecessary tasks being run in each iteration! This is highly undesirable in situations where the workers are utilized across multiple other jobs besides running the iterative algorithm mentioned above.

The goal of our work is to design new task allocation strategies, with rigorous theoretical support, that would attempt

to minimize the expected completion time subject to the *constraint* that such wastefulness is completely eliminated. That is, we ensure that no more than $B$ tasks are completed in each round.

### 1.1. A Motivating Example: Optimal Parallel SGD

A key inspiration for our work, and the prime example of the general task collection problem described above, relates to recent development in the area of parallel stochastic gradient descent (SGD) methods. Consider the problem of finding an approximate stationary point of the optimization problem

$$\min_{\boldsymbol{x} \in \mathbb{R}^d} \left\{ f(\boldsymbol{x}) := \mathbb{E}_{\boldsymbol{\xi} \sim \mathcal{D}} \left[ f_{\boldsymbol{\xi}}(\boldsymbol{x}) \right] \right\},$$

where $f_{\boldsymbol{\xi}} : \mathbb{R}^d \to \mathbb{R}$ are smooth nonconvex functions, and $f$ is assumed to be bounded from below. We assume that

$$\mathbb{E}_{\boldsymbol{\xi} \sim \mathcal{D}} \left[ \| f_{\boldsymbol{\xi}}(\boldsymbol{x}) - \nabla f(\boldsymbol{x}) \|^2 \right] \leq \sigma^2$$

for all $\boldsymbol{x} \in \mathbb{R}^d$.

In a recent breakthrough, Tyurin & Richtárik (2024) recently developed a parallel SGD method, *optimal* in terms of a novel notion of complexity called *time complexity*, for solving the above problem with $n$ parallel workers, assuming that it takes $\tau_i > 0$ seconds to worker $i$ to compute a stochastic gradient of $f$ (this corresponds to a task). Their method, Rennala SGD, corresponds to Minibatch SGD of minibatch size $B$ (which depends on the target accuracy and $\sigma$ only), with the $B$ tasks (stochastic gradients) completed via GTA. While minimax optimal in terms of time complexity, the GTA task allocation strategy employed within Rennala SGD can be wasteful, as explained above.

Recently, Maranjyan et al. (2025b) proposed Ringmaster ASGD, a fully asynchronous SGD method, matching the optimal time complexity of Rennala SGD and achieving optimality for arbitrary compute time patterns associated with the tasks (stochastic gradients), including random, as considered in our setup. However, Ringmaster ASGD also employs a greedy task allocation strategy, leading to wastefulness.

Numerous other parallel/distributed methods involve the implementation of a task allocation strategy, including stochastic proximal point methods (task = evaluation of the stochastic prox operator), higher-order methods (task = evaluation of stochastic Hessian), and beyond. So, by addressing the general task allocation problem, we aim to tame the inherent resource wastefulness of all these methods.

### 1.2. Contributions

In this work, we formalize the task allocation problem as a *combinatorial online learning problem with partial feedback and non-linear losses*. Then, we introduce ATA, a lower-confidence bound-based algorithm designed to solve the proposed allocation problem. ATA is agnostic to workers' computation times, and our theoretical analysis demonstrates that the total computation time achieved by our methods remains within a small multiplicative factor of the optimal computation time (i.e., the one attainable with full knowledge of the workers' arm distributions). Additionally, we present ATA-Empirical, a variant of ATA that leverages a novel data-dependent concentration inequality and achieves better empirical results. Finally, we validate our approach through numerical simulations.

## 2. Related Work

Most of the literature on asynchronous methods focuses on demonstrating advantages over their synchronous counterparts. For the simplest method, SGD, this was only recently established by Tyurin & Richtárik (2024). With this result in place, the community can now shift its focus to reducing the overhead of asynchrony. Our work may be the first step in this direction.

In federated learning (FL) (Konečný et al., 2016; McMahan et al., 2016; Kairouz et al., 2021), several works account for system heterogeneity. The most well-known FL method, FedAvg (McMahan et al., 2017), operates by performing multiple local steps on workers, where each step can be viewed as a task. Some works adjust the number of local steps based on worker computation times (Li et al., 2020; Maranjyan et al., 2022), effectively adapting task assignments to worker speed. However, these methods rely on prior knowledge of these times rather than learning them adaptively, as we do.

We reformulate our problem as an online bandit problem. The literature on bandit algorithms is vast, and we refer the reader to Lattimore & Szepesvári (2020) for an introduction to this subject. Our algorithm is based on the approach of using Lower Confidence Bounds (LCBs) on the true means of the arms. This idea, originally proposed by Auer (2002) for the classical Multi-Armed Bandit (MAB) setting, has since been widely adopted in the stochastic combinatorial bandits literature (Gai et al., 2012; Chen et al., 2013; Combes et al., 2015; Kveton et al., 2015). Using LCBs instead of the empirical estimates of the means allows to trade-off optimally exploration and exploitation.

The "greedy" approach we employ, which involves selecting the action that minimizes the loss function based on lower confidence bounds instead of the unknown means, is a standard technique in the literature (Chen et al., 2013; Lin et al., 2015). However, note that our larger action space and the discontinuity of our loss function necessitates a more tailored analysis. To the best of our knowledge, this is the first work addressing a non-continuous loss function in a stochastic combinatorial MAB-like framework. To over-

come this challenge, we exploit the specific structures of our loss function and action space to control the number of rounds where suboptimal actions are chosen. Additionally, our procedure is computationally efficient.

## 3. Problem Setup

In this section, we formally describe the problem setup.

### 3.1. Task Allocation Protocol

We consider a system of $n$ workers, each responsible for computing gradients. In each round, the allocation algorithm has a budget of $B$ units that must be allocated among the $n$ workers. Each unit allocation will result in one gradient computation. We denote by $K$ the total number of rounds, which is assumed to be unknown to the learner. We denote by $X_{i,k}^{(u)}$ the computation time of the worker $i \in [n] := \{1, 2, \ldots, n\}$ for round $k \in [K]$ on its $u$-th gradient. Consequently, the computation time required for worker $i$ to perform its task of computing $a_{i,k}$ gradients in round $k$ is given by

$$\sum_{u=1}^{a_{i,k}} X_{i,k}^{(u)}$$

if $a_{i,k} \geq 1$, and 0 otherwise.

In each round $k$, the allocation algorithm must choose an allocation vector $\boldsymbol{a}_k \in \mathbb{N}^n$ such that $\|\boldsymbol{a}\|_1 = B$, based on the information available prior to round $k$. The feedback consists of $a_{i,k}$ observed times for all the chosen workers. We will denote the action set by

$$\mathcal{A} := \{\boldsymbol{a} \in \mathbb{N}^n : \|\boldsymbol{a}\|_1 = B\},$$

where $\mathbb{N}$ is the set of natural numbers, including the 0.

The objective of the allocation strategy in each round $k$ is to minimize the total computation time. Hence, the objective is to minimize $C : \mathcal{A} \to \mathbb{R}_+$, the computation time that the optimizer waits to receive $B$ gradients using an allocation vector $\boldsymbol{a} \in \mathcal{A}$, defined as

$$C(\boldsymbol{a}_k) := \max_{i \in \operatorname{supp}(\boldsymbol{a}_k)} \sum_{u=1}^{a_{i,k}} X_{i,k}^{(u)} . \quad (1)$$

### 3.2. Modeling Assumptions

We assume that the computation time of each worker $i \in [n]$ are i.i.d. drawn from a random variable $X_i$ following a probability distribution $\nu_i$. We denote by $\boldsymbol{\mu} = (\mu_1, \ldots, \mu_n)$ the vector of unknown means. Hence, the random variables $(X_{i,k}^{(u)})$ with $u \in \{1, \ldots, a_{i,k}\}$ are $a_{i,k}$ i.i.d. samples drawn from $\nu_i$.

We assume that the distribution $\nu_i$ of the computation times to be sub-exponential random variables. To quantify this

assumption, we recall the definition of the sub-exponential norm, also known as the Orlicz norm, for a centered real-valued random variable $X$:

$$\|X\|_{\psi_1} := \inf\{C > 0 : \mathbb{E}[\exp(|X|/C)] \leq 2\} . \quad (2)$$

Hence, formally we make the following assumption.

**Assumption 3.1.** Let $\alpha \geq 0$. For all $i \in [n]$, $X_i$ is a positive random variable and $\|X_i - \mu_i\|_{\psi_1} \leq \alpha$.

In the remainder of this paper we denote $\alpha_i := \|X_i\|_{\psi_1}$ for each $i \in [n]$, let $\alpha := \max_{i \in [n]} \alpha_i$.

The considered class encompasses several other well-known classes of distributions in the literature, such as support-bounded and sub-Gaussian distributions. Moreover, it includes exponential distributions, which are frequently used in the literature to model waiting or computation times in queuing theory and resource allocation in large distributed systems (Gelenbe & Mitrani, 2010; Gross et al., 2011; Hadjis et al., 2016; Mitliagkas et al., 2016; Dutta et al., 2018; Nguyen et al., 2022).

### 3.3. Objective of the Allocation Algorithm

The main objective of this work is to develop an online allocation strategy with small expected total computation time, defined as

$$\mathcal{C}_K := \sum_{k=1}^{K} \mathbb{E}[C(\boldsymbol{a}_k)].$$

If the distributions of the arms were known in advance, the optimal allocation $\boldsymbol{a}^* \in \mathcal{A}$ would be selected to minimize the expected computation time per round, $\mathbb{E}[C(\cdot)]$, and this allocation would be used consistently over $K$ rounds, leading to the optimal total computation time

$$\mathcal{C}_K^* = K\mathbb{E}[C(\boldsymbol{a}^*)].$$

Our goal is to design a strategy that ensures the computation time $\mathcal{C}_K$ remains within a small multiplicative factor of the optimal time $\mathcal{C}_K^*$, plus an additional negligible term. Specifically, we aim to satisfy

$$\mathcal{C}_K \leq \gamma \cdot \mathcal{C}_K^* + \mathcal{E}_K, \quad (3)$$

where $\gamma \geq 1$ is a constant close to 1, and $\mathcal{E}_K$ is a negligible term compared to $\mathcal{C}_K^*$ when $K \to \infty$. This would assure us that in the limit we are a constant multiplicative factor away from the performance of the optimal allocation strategy that has full knowledge of the distributions of the computational times of the workers.

Finding a strategy solving the objective in (3) presents several technical challenges. First, the action space $\mathcal{A}$ is discrete, and the nonlinearity of the computation time functions

$C(\cdot)$ prevents reducing our objective to a convex problem. Second, the size of $\mathcal{A}$ is combinatorial, growing on the order of $\binom{n+B-1}{B}$, which necessitates exploiting the inherent problem structure to develop efficient strategies. Third, because the workers' computation times are stochastic, any solution must account for uncertainty. Finally, the online setting forces the learner to balance exploration and exploitation under a limited allocation budget of $B$ units per round and partial feedback—only the computation times of workers who receive allocations are observed. This last point naturally suggests adopting a MAB approach.

In the next section, we show how to reduce this problem to a MAB problem and how to efficiently solve it.

# 4. Adaptive Task Allocation

Here, we first show how to reduce the problem in (3) to a *non-linear* stochastic Multi-Armed Bandit (MAB) problem. Then, we propose an efficient algorithm for this formulation.

## 4.1. Reduction to Multi-Armed Bandit and Proxy Loss

The stochastic MAB problem is a fundamental framework in sequential decision-making under uncertainty. It involves a scenario where an agent must choose among a set of arms, each associated with an unknown reward distribution. The agent aims to maximize cumulative reward (or equivalently minimize the cumulative loss) over time by balancing exploration (gathering information about the reward distributions) and exploitation (leveraging the best-known arm). The challenge lies in the trade-off between exploring suboptimal arms to refine reward estimates and exploiting the arm with the highest observed reward, given the stochastic nature of the outcomes. Using the terminology from bandit literature, here we will refer to each worker as an "arm."

However, differently from the standard MAB problem, we have a harder problem because $\mathbb{E}\left[C(\boldsymbol{a}_k)\right]$ depends on the joint distribution of all the arms in the support of $\boldsymbol{a}_k$, rather than on their expectations only. This dependency potentially renders the task of relying on estimates of $\mathbb{E}\left[C(\boldsymbol{a})\right]$ for $\boldsymbol{a} \in \mathcal{A}$ computationally challenging due to the combinatorial nature of the set $\mathcal{A}$.

To solve this issue, our first idea is to introduce a *proxy loss* $\ell : \mathcal{A} \times \mathbb{R}_{\geq 0}^n \to \mathbb{R}_{\geq 0}$, defined as

$$\ell(\boldsymbol{a}, \boldsymbol{\mu}) := \max_{i \in [n]} a_i \mu_i . \quad (4)$$

Due to the convexity of $C(\cdot)$, the introduced proxy-loss underestimates the expected computation time. However, in Appendix D.3 we prove that this quantity also upper bounds the expected computation time up to a constant that depends on the distribution of the arms. In particular, for any $\boldsymbol{a} \in \mathcal{A}$,

we show that

$$\ell(\boldsymbol{a}, \boldsymbol{\mu}) \leq \mathbb{E}\left[C(\boldsymbol{a})\right] \leq (1 + 4\eta \ln(B))\ell(\boldsymbol{a}, \boldsymbol{\mu}), \quad (5)$$

where $\eta$ is defined as

$$\eta := \max_{i \in [n]} \frac{\alpha_i}{\mu_i} . \quad (6)$$

In words, $\eta$ provides an upper bound on the ratio between the standard deviation and the mean of the arms. Note that in the literature, it is common to consider exponential, Erlang, or Gamma distributions, where the ratio $\eta$ is typically[1] bounded by $1$.

The bound above will allow us to derive guarantees on the total computation time of an allocation strategy based on its guarantees for the proxy loss $\ell(\cdot)$, up to a factor of the order $1 + 4\eta \ln(B)$. We remark that in the special case where the arms' distributions are deterministic ($\eta = 0$) or the query budget is unitary ($B = 1$), the two targets $\mathbb{E}\left[C(\boldsymbol{a})\right]$ and $\ell$ exactly coincide.

## 4.2. Comparison with the Combinatorial Bandits Setting

Our setting is closely related to the Combinatorial Multi-Armed Bandits (CMAB) framework (Cesa-Bianchi & Lugosi, 2012), particularly due to the combinatorial nature of the action space and the semi-bandit feedback, where the learner observes outcomes from all chosen arms. However, our formulation differs in two significant ways. First, while CMAB typically involves selecting a subset of $n$ arms, resulting in an action space with a maximum size of $2^n$, our action space $\mathcal{A}$ has a cardinality of $\binom{n+B-1}{B}$. The ratio between these two can be extremely large, potentially growing exponentially with $n$. Second, although most works in this domain assume a linear loss function in the arms' means, some notable exceptions address non-linear reward functions (Chen et al., 2013; Lin et al., 2015; Chen et al., 2016b; Wang & Chen, 2018). However, these approaches generally rely on assumptions such as smoothness, Lipschitz continuity, or higher-order differentiability of the reward function. In contrast, our loss function $\ell(\cdot, \boldsymbol{\mu})$ is not continuous with respect to the arms' means. Finally, motivated by the practical requirements of our setting, we place a strong emphasis on computational efficiency that rules out most of the approaches based on CMAB.

## 4.3. Adaptive Task Allocation Algorithm

Now, we introduce our Adaptive Task Allocation algorithm (ATA). ATA does not require prior knowledge of the horizon $K$ and only relies on an upper bound $\alpha$ satisfying

---

[1]For $\mathrm{Gamma}(\alpha, \lambda)$, $\sigma/\mu = 1/\sqrt{\alpha}$, so the claim holds for $\alpha \geq 1$.

**Algorithm 1** ATA (Adaptive Task Allocation)

1: **Input**: allocation budget $B$, $\alpha > 0$
2: **Initialize**: empirical means $\hat{\mu}_{i,1} = 0$, usage counts $K_{i,1} = 0$, and usage times $T_{i,1} = 0$, for all $i \in [n]$
3: **for** $k = 1, \ldots, K$ **do**
4:      Compute LCBs $(s_{i,k})$ for all $i \in [n]$ using (7)
5:      Find allocation: $\boldsymbol{a}_k \in \arg\min_{\boldsymbol{a} \in \mathcal{A}} \ell(\boldsymbol{a}, \boldsymbol{s}_k)$
6:      Allocate $a_{i,k}$ tasks to each worker $i \in [n]$
7:      Update optimization parameters
8:      **for** $i$ such that $a_{i,k} \neq 0$ **do**
9:          $K_{i,k+1} = K_{i,k} + a_{i,k}$
10:         $T_{i,k+1} = T_{i,k} + \sum_{j=1}^{a_{i,k}} X_{i,k}^{(j)}$
11:         $\hat{\mu}_{i,k+1} = T_{i,k+1}/K_{i,k+1}$
12:      **end for**
13: **end for**

---

$\alpha \geq \max_{i \in [n]} \|X_i - \mu_i\|_{\psi_1}$ for the Orlicz norms of the arm distributions. Recall that $\|X_i - \mu_i\|_{\psi_1} \leq 2 \|X_i\|_{\psi_1}$, so an upper bound on $\|X_i\|_{\psi_1}$ also provides one for $\|X_i - \mu_i\|_{\psi_1}$. The core idea of the procedure is to allocate the workers based on *lower confidence bound estimates* on the arm means $(\mu_i)_{i \in [n]}$, in order to balance exploration and exploitation.

For each arm $i \in [n]$ and round $k \in [K]$, let $K_{i,k}$ represent the number of samples collected from the distribution of arm $i$ up to round $k$. At each round $k$, we compute an empirical mean, denoted by $\hat{\mu}_{i,k}$, using the $K_{i,k}$ samples obtained so far. Based on these empirical means, we define the lower confidence bounds $s_{i,k}$ as

$$s_{i,k} = (\hat{\mu}_{i,k} - \text{conf}(i,k))_+ , \qquad (7)$$

where $(x)_+ = \max\{x, 0\}$ and $\text{conf}(\cdot, \cdot)$ is defined as

$$\text{conf}(i,k) = \begin{cases} 2\alpha \left( \sqrt{\frac{\ln(2k^2)}{K_{i,k}}} + \frac{\ln(2k^2)}{K_{i,k}} \right), & K_{i,k} \geq 1, \\ +\infty, & K_{i,k} = 0 . \end{cases}$$

The term $\text{conf}(\cdot, \cdot)$ is derived from a known concentration inequality for sub-exponential variables with an Orlicz norm bounded by $\alpha$ (Lemma E.1 in the Appendix).

Given the confidence bounds $\boldsymbol{s}_k := (s_{1,k}, \ldots, s_{n,k})$, the learner selects the action $\boldsymbol{a}_k \in \mathcal{A}$ at round $k$ that minimizes the loss $\ell(\cdot, \boldsymbol{s}_k)$, defined in (4). While nonconvex, we show in Appendix C that this optimization problem can be solved using a recursive routine, whose computational efficiency is $\mathcal{O}(n \ln(\min\{B, n\}) + \min\{B, n\}^2)$.

*Remark* 4.1. Line 7 of the algorithm acts as a placeholder for the optimization method, where the optimization parameters are updated using the quantities computed by the workers (e.g., gradients in the case of SGD). In this view, the allocation algorithm is independent of the specifics of

the chosen optimization algorithm. Refer to Appendix B for further details.

As last step, the feedback obtained after applying the allocation $\boldsymbol{a}_k$ is used to update the lower confidence bounds. The complete pseudocode for ATA is provided in Algorithm 1.

### 4.4. Upper-Bound on the Total Computation Time

We provide guarantees for ATA in the form of an upper bound on the expected total computation time required to perform $K$ iterations of the optimization procedure. Recall that the proxy loss $\ell(\cdot, \boldsymbol{\mu})$ and the expected computation time are related through (5). This relationship and Theorem 6.1 allow us to derive guarantees on the expected total computation time, denoted by

$$\mathcal{C}_K := \sum_{k=1}^{K} \mathbb{E}\left[C(\boldsymbol{a}_k)\right] .$$

We define the optimal allocation for minimizing the computation time as

$$\boldsymbol{a}^* \in \arg\min_{\boldsymbol{a} \in \mathcal{A}} \mathbb{E}\left[C(\boldsymbol{a})\right] .$$

Consequently, the optimal expected total computation time in this framework is given by

$$\mathcal{C}_K^* := K \mathbb{E}\left[C(\boldsymbol{a}^*)\right] .$$

**Theorem 4.2** (Proof in Appendix D.3). *Suppose Assumption 3.1 holds and let $\eta := \max_{i \in [n]} \alpha_i / \mu_i$. Then, the total expected computation time after $K$ rounds, using the allocation prescribed by* ATA *with inputs* $(B, \alpha)$ *satisfies*

$$\mathcal{C}_K \leq (1 + 4\eta \ln(B)) \mathcal{C}_K^* + \mathcal{O}(\ln K) .$$

*Remark* 4.3. The $\mathcal{O}(\cdot)$ term hides an instance dependent factor. We will give its full specifics in the regret upper bound of Theorem 6.1.

The bound in Theorem 4.2 shows that the total expected computation time of ATA remains within a multiplicative factor of $1 + 4\eta \ln(B)$ of the optimal computation time $\mathcal{C}_K^*$, with an additional remainder term that scales logarithmically with $K$. Since $\mathcal{C}_K = \Omega(K)$, this additive term is negligible compared to $\mathcal{C}_K^*$. In practical scenarios, where computation time follows common distributions such as exponential or Gamma, the factor $\eta$ is typically of order 1, and $\ln(B)$ remains relatively small for the batch sizes commonly used in optimization algorithms like SGD.

The reader might wonder if the more ambitious goal of deriving bounds with a multiplicative factor of exactly 1 is achievable. However, achieving this goal would require significantly more precise estimates of the expected computation time $\mathbb{E}[C(\boldsymbol{a})]$ for all $\boldsymbol{a} \in \mathcal{A}$. Since $\mathbb{E}[C(\boldsymbol{a})]$ depends

on the joint distribution of all workers in the support $\boldsymbol{a}$, obtaining such precise estimates would come at the cost of computational efficiency in the allocation strategy.

We note that it is unsurprising that $\eta$ appears in the upper bound of Theorem 4.2, since having a heavier-tailed distribution increases the gap between $\ell(\boldsymbol{a}, \boldsymbol{\mu})$ and $\mathbb{E}\left[C(\boldsymbol{a})\right]$ through the convexity of $C(\cdot)$. Instead, the factor $\ln(B)$ arises because $C(\cdot)$ is expressed as the maximum of up to $B$ random variables. Moreover, in the edge cases where $\eta = 0$ (deterministic case) or $B = 1$ (linear cost function), we guarantee that the expected computation time is at most an *additive* factor away from the optimal one.

## 5. Empirical Adaptive Task Allocation

The ATA procedure is based on a lower confidence bound approach that relies on concentration inequalities. These bounds play a key role in performance, as sharper concentration bounds lead to more accurate estimates and reduce exploration of suboptimal options. Since workers' computation times follow sub-exponential distributions, their concentration behavior is determined by the Orlicz norm of the corresponding variables. In ATA, the only prior knowledge available is an upper bound on the largest Orlicz norm among all arms. When the Orlicz norms of the arms' distributions vary significantly, this uniform bound may result in loose confidence intervals and inefficient exploration.

To address this issue, we introduce ATA-Empirical, which better adapts to the distribution of each arm, particularly its Orlicz norm. This adaptation is achieved through a novel data-dependent concentration inequality for sub-exponential variables. Unlike ATA, which depends on the maximum Orlicz norm, ATA-Empirical accounts for the individual Orlicz norms of all arms, denoted by $(\alpha_i)_{i \in [n]}$. This improvement is reflected in the upper bounds on regret presented in Section 6. In practice, this leads to improved performance at least some settings, as shown in our simulations in Section 7. However, this increased adaptivity comes with a trade-off since ATA-Empirical requires an upper bound on the quantity $\eta = \max_i \alpha_i/\mu_i$, rather than a bound on the largest Orlicz norm. That said, for many distributions of interest, the ratios $\alpha_i/\mu_i$ across different arms tend to be of the same order, whereas their Orlicz norms can vary significantly.

The ATA-Empirical procedure differs from ATA only in the lower confidence bounds it uses. These bounds are derived from the novel concentration inequality in Lemma 6.2 and are defined for arm $i \in [n]$ at round $k \in [K]$ as

$$\hat{s}_{i,k} = \hat{\mu}_{i,k}\left[1 - 2\eta\left(\sqrt{\frac{\ln(2k^2)}{K_{i,k}}} + \frac{\ln(2k^2)}{K_{i,k}}\right)\right]_+, \quad (8)$$

where $\eta = \max_{i \in [n]} \alpha_i/\mu_i$.

The expected total computation time $\mathcal{C}_K$ of ATA-Empirical satisfies the same guarantee presented in Theorem 6.1, but we obtain an improved multiplicative factor of the additive logarithmic term. The precise expressions of these factors are provided in the next section, and they show that the guarantees of ATA-Empirical adapt to the Orlicz norms $\|X_i\|_{\psi_1}$ of each arm, while the guarantees of ATA depend on the maximum Orlicz norm $\max_i \|X_i\|_{\psi_1}$.

## 6. Theoretical Results

In this section, we sketch the derivation of Theorem 4.2 for ATA and ATA-Empirical, through a regret analysis on the proxy losses. We define the expected cumulative regret of the proxy loss $\ell(\cdot, \boldsymbol{\mu})$ after $K$ rounds

$$\mathcal{R}_K := \sum_{k=1}^{K} \mathbb{E}\left[\ell(\boldsymbol{a}_k, \boldsymbol{\mu})\right] - K \cdot \ell(\bar{\boldsymbol{a}}, \boldsymbol{\mu}), \quad (9)$$

where $\bar{\boldsymbol{a}} \in \arg\min_{\boldsymbol{a} \in \mathcal{A}} \ell(\boldsymbol{a}, \boldsymbol{\mu})$ represents the optimal allocation over the workers. If multiple optimal actions exist, we consider the one returned by the optimization sub-routine used in ATA (line 5 of Algorithm 2).

We derive upper bounds on the expected cumulative regret $\mathcal{R}_K$. Based on these bounds, we provide the guarantees on the expected total computation time required to complete $K$ iterations of the optimization process.

### 6.1. Guarantees for ATA

For each worker $i \in [n]$, recall that $\bar{a}_i$ denote the prescribed allocation of the optimal action $\bar{\boldsymbol{a}}$. Define $k_i$ as the smallest integer satisfying

$$(\bar{a}_i + k_i)\mu_i > \ell(\bar{\boldsymbol{a}}, \boldsymbol{\mu}) . \quad (10)$$

From the definition above, it follows that if the learner plays an action $\boldsymbol{a}_k$ at round $k$ such that $a_{k,i} \geq \bar{a}_i + k_i$, then $\ell(\boldsymbol{a}_k, \boldsymbol{\mu}) \geq \ell(\bar{\boldsymbol{a}}, \boldsymbol{\mu})$. Thus, $k_i$ can be interpreted as the smallest number of additional units allocated to worker $i$ that result in a suboptimal loss. Moreover, for every worker $i \in [n]$, we have $k_i \in \{1, 2\}$ (see Lemma D.1 in the Appendix).

The next result provides an upper bound on the expected regret of ATA.

**Theorem 6.1** (Proof in Appendix D.1). *Suppose that Assumption 3.1 holds. Then, the expected regret of ATA with inputs $(B, \alpha)$ satisfies*

$$\mathcal{R}_K \leq 2n \max_{i \in [n]}\{B\mu_i - \ell(\bar{\boldsymbol{a}}, \boldsymbol{\mu})\}$$

$$+ c \cdot \sum_{i=1}^{n} \frac{\alpha^2(\bar{a}_i + k_i)(B\mu_i - \ell(\bar{\boldsymbol{a}}, \boldsymbol{\mu}))}{((\bar{a}_i + k_i)\mu_i - \ell(\bar{\boldsymbol{a}}, \boldsymbol{\mu}))^2} \cdot \ln K,$$

*where $\alpha \geq \max_{i \in [n]} \|X_i - \mu_i\|_{\psi_1}$, and $c$ is a numerical constant.*

The first term in the regret upper bound is independent on the number of rounds $K$. The second term, however, grows logarithmically with $K$, which aligns with the behavior observed in stochastic bandit problems in the literature.

In the case where $B = 1$, our setting reduces to the problem of regret minimization for the standard multi-armed bandits. Observe that in this case $\ell(\bar{\boldsymbol{a}}, \boldsymbol{\mu}) = \min_{i \in [n]} \mu_i$, $k_i = 1$ for all $i \in [n]$. Therefore, the guarantees of Theorem 6.1 recover the known optimal bound

$$\mathcal{O}\left(\sum_i \ln(K)/\Delta_i\right)$$

of the standard MAB setting, where $\Delta_i := \mu_i - \min_j \mu_j$.

**Proof sketch.** In standard and combinatorial MAB problems, regret bounds are typically derived by controlling the number of rounds in which the learner selects suboptimal arms. These bounds are often of the order $\ln(K)/\Delta^2$, where $\Delta$ denotes the suboptimality gap and quantifies the exploration cost required to distinguish optimal actions from suboptimal ones.

In our setting, the problem is more complex since the learner must not only choose which arms to pull but also determine the allocation of resources across selected arms. With this in mind, we develop the following key arguments leading to the bound in Theorem 6.1.

We define *over-allocation* for worker $i$ at round $k$ as the event where $a_{i,k} \geq \bar{a}_i + k_i$. By definition of $k_i$ (see (10)), this implies that $\ell(\boldsymbol{a}_k, \boldsymbol{\mu}) > \ell(\bar{\boldsymbol{a}}, \boldsymbol{\mu})$. We define a *bad round* as a round where $\ell(\boldsymbol{a}_k, \boldsymbol{\mu}) > \ell(\bar{\boldsymbol{a}}, \boldsymbol{\mu})$, and we say that a bad round is *triggered by arm* $i$ when $a_{i,k}\mu_i = \ell(\boldsymbol{a}_k, \boldsymbol{\mu}) > \ell(\bar{\boldsymbol{a}}, \boldsymbol{\mu})$. Then, the proof revolves around establishing an upper bound on the total number of bad rounds.

To derive this bound, we consider the number of samples required to verify that the mean computation time of worker $i$ under over-allocation exceeds the optimal waiting time $\ell(\bar{\boldsymbol{a}}, \boldsymbol{\mu})$. Specifically, we need to test whether the mean of the corresponding distribution, at least $(\bar{a}_i + k_i)\mu_i$, surpasses the threshold $\ell(\bar{\boldsymbol{a}}, \boldsymbol{\mu})$. This is equivalent to testing whether

$$\left\{\mu_i > \frac{\ell(\bar{\boldsymbol{a}}, \boldsymbol{\mu})}{\bar{a}_i + k_i}\right\}.$$

Using the concentration inequality applied in our analysis, the number of samples required for this test is of the order:

$$\alpha_i^2\left(\mu_i - \frac{\ell(\bar{\boldsymbol{a}}, \boldsymbol{\mu})}{\bar{a}_i + k_i}\right)^{-2} = \frac{\alpha_i^2(\bar{a}_i + k_i)^2}{((\bar{a}_i + k_i)\mu_i - \ell(\bar{\boldsymbol{a}}, \boldsymbol{\mu}))^2}. \quad (11)$$

During rounds where worker $i$ is over-allocated, the learner collects at least $\bar{a}_i + k_i$ samples from the corresponding distribution. Therefore, the total number of rounds required to accumulate enough samples to stop over-allocating worker $i$ can be upper-bounded by

$$\frac{\alpha_i^2(\bar{a}_i + k_i)}{((\bar{a}_i + k_i)\mu_i - \ell(\bar{\boldsymbol{a}}, \boldsymbol{\mu}))^2}.$$

In the regret bound of Theorem 6.1, the term $\alpha^2$ appears instead of $\alpha_i^2$ because the learner's prior knowledge is limited to an upper bound $\alpha \geq \max_i \|X_i - \mu_i\|_{\psi_1}$ on the maximal Orlicz norm of the arm distributions. Finally, considering that the worst-case excess loss incurred when over-allocating worker $i$ is $B\mu_i - \ell(\bar{\boldsymbol{a}}, \boldsymbol{\mu})$, we obtain the stated bound.

### 6.2. Guarantees for ATA-Empirical

We present theoretical guarantees for ATA-Empirical by providing an upper bound on the expected cumulative regret (9). As discussed in Section 4, ATA-Empirical leverages lower confidence bounds derived from a novel data-dependent concentration inequality introduced below. The proof of this result is detailed in Appendix E.

**Lemma 6.2.** *Let $X_1, \ldots, X_n$ be i.i.d. positive random variables with mean $\mu$, such that $\alpha = \|X_1 - \mu\|_{\psi_1} < +\infty$. Let $\hat{X}_n$ denote the empirical mean. For $\delta \in (0, 1)$, let*

$$C_{n,\delta} := 2\sqrt{\frac{\log(2/\delta)}{n}} + 2\frac{\log(2/\delta)}{n},$$

*where $\eta = \alpha/\mu$. Then, with probability at least $1 - \delta$, we have*

$$\mu \geq \hat{X}_n\left(1 - \eta C_{n,\delta}\right)_+.$$

*Moreover, if $\eta C_{n,\delta} \leq \frac{1}{4}$, then, we have with probability at least $1 - \delta$, we have*

$$\hat{X}_n\left(1 - \eta C_{n,\delta}\right)_+ \leq \mu \leq \hat{X}_n\left(1 + \frac{4}{3}\eta C_{n,\delta}\right).$$

Using the concentration inequality above, we construct the lower confidence bounds $\hat{s}_{i,k}$ as defined in (8). The following theorem provides an upper bound on the regret of ATA-Empirical.

**Theorem 6.3** (Proof in Appendix D.2). *Suppose that Assumption 3.1 holds. Then, the expected regret of ATA-Empirical with inputs $(B, \eta)$, satisfies*

$$\mathcal{R}_K \leq 2n \max_{i \in [n]}\{B\mu_i - \ell(\bar{\boldsymbol{a}}, \boldsymbol{\mu})\}$$

$$+ c\eta^2 \cdot \sum_{i=1}^{n} \frac{\mu_i^2(\bar{a}_i + k_i)(B\mu_i - \ell(\bar{\boldsymbol{a}}, \boldsymbol{\mu}))}{((\bar{a}_i + k_i)\mu_i - \ell(\bar{\boldsymbol{a}}, \boldsymbol{\mu}))^2} \cdot \ln K,$$

*where $\eta \geq \max_{i \in [n]} \alpha_i/\mu_i$ and $c$ is a numerical constant.*

Comparing the bounds for ATA-Empirical and ATA, we observe a key differences. Unlike the bound in Theorem 6.1, which incurs a squared maximal Orlicz norm penalty of $\alpha^2$ for all terms in the upper bound, ATA-Empirical benefits from its adaptive nature, leading to a term-specific factor of $\eta^2 \mu_i^2$. In the case where the arm distributions have a ratio $\alpha_i / \mu_i$ of the same order (such as the exponential distributions), the bound of Theorem 6.3 shows that ATA-Empirical adapts to the quantities $\alpha_i$ as we have, in the last case, $\eta \mu_i = \alpha_i$.

## 7. Experiments

In this section, we validate our algorithms by simulating a scenario with $n$ workers, where we solve a simple problem using SGD. In each iteration, we collect $B = 23$ gradients from the workers and perform a gradient descent step.

The objective function $f : \mathbb{R}^d \to \mathbb{R}$ is a convex quadratic defined as

$$f(x) = \frac{1}{2} \boldsymbol{x}^\top \mathbf{A} \boldsymbol{x} - \boldsymbol{b}^\top \boldsymbol{x},$$

where

$$\mathbf{A} = \frac{1}{4} \begin{bmatrix} 2 & -1 & & 0 \\ -1 & \ddots & \ddots & \\ & \ddots & \ddots & -1 \\ 0 & & -1 & 2 \end{bmatrix} \in \mathbb{R}^{d \times d},$$

$$\boldsymbol{b} = \frac{1}{4} \begin{bmatrix} -1 \\ 0 \\ \vdots \\ 0 \end{bmatrix} \in \mathbb{R}^d.$$

We denote $f^*$ as the minimum value of the function $f$. Each of the $n$ workers is able to calculate unbiased stochastic gradients $\boldsymbol{g}(\boldsymbol{x})$ that satisfy

$$\mathbb{E}\left[\|\boldsymbol{g}(\boldsymbol{x}) - \nabla f(\boldsymbol{x})\|^2\right] \le 0.01^2.$$

This is achieved by adding Gaussian noise to the gradients of $f$.

The computation time for worker $i$ is modeled by the distribution

$$\nu_i = 29\sqrt{i} + \mathrm{Exp}\left(29\sqrt{i}\right),$$

for all $i \in [n]$, where $\mathrm{Exp}(\beta)$ denotes the exponential distribution with scale parameter $\beta$. The expected value of this distribution is $\mu_i = 2 \cdot 29\sqrt{i}$. Furthermore, the Orlicz norm satisfies the bound $\alpha_i \le 2\mu_i$.

We consider three benchmark algorithms. GTA-SGD, originally introduced as Rennala SGD by Tyurin & Richtárik (2024). Additionally, we include OFTA (Optimal Fixed Task Allocation), which assumes the oracle knowledge of the mean computation times and uses the optimal allocation $\bar{\boldsymbol{a}}$

in (9) in each iteration, and UTA (Uniform Task Allocation), which distributes $B$ tasks uniformly among the $n$ workers. If $n > B$, then in UTA we select $B$ workers at random, each one tasked to calculate one stochastic gradient. Our algorithms aim to achieve a performance close to the one of OFTA, without any prior knowledge of the true means.

For ATA we set $\alpha = \alpha_n = 4 \cdot 29\sqrt{n}$, while for ATA-Empirical we use $\eta = 1$. The results of our experiments are shown in Figure 1. As expected, GTA is the fastest in terms of runtime (first column), but it performs poorly in terms of total worker time (second column). This is because it uses all devices, most of which perform useless computations that are never used, leading to worse performance as the number of workers increases. In fact, its performance can become arbitrarily worse. On the other hand, OFTA performs best in terms of total worker time. Although it is slower in terms of runtime, the difference is by a constant factor that does not increase as $n$ grows. This is because additional workers are less efficient and do not provide significant benefits for GTA.

Turning our attention to our algorithms, both ATA and ATA-Empirical initially behave like UTA, as it is expected by the need to perform an initial exploration phase with uniform allocations. However, after this phase, they begin to converge to the performance of OFTA.

The last two columns contain plots that confirm our theoretical derivations. The third plot validates Theorem 4.2, showing that ATA and ATA-Empirical converge to OFTA up to a constant. The final column shows the averaged cumulative regret, vanishing over time as predicted by Theorems 6.1 and 6.3.

Table 1: Ratios of total worker times and runtimes required to achieve $f(\boldsymbol{x}) - f^* < 10^{-5}$. For total worker time, we divide the total worker time of GTA by the corresponding total worker times of the other algorithms listed. For runtime, we do the opposite, dividing the runtime of the other algorithms by the runtime of GTA, since GTA is the fastest. To simplify the naming, we refer to ATA-Empirical as ATA-E.

| $n$ | TOT. WORKER TIME RATIO | | | RUNTIME RATIO | | |
| --- | --- | --- | --- | --- | --- | --- |
| | ATA | ATA-E | OFTA | ATA | ATA-E | OFTA |
| 17 | 1.3 | 1.26 | 1.26 | 1.73 | 1.75 | 1.74 |
| 51 | 2.91 | 2.69 | 3.03 | 2.43 | 2.45 | 2.17 |
| 153 | 7.22 | 7.02 | 9.1 | 3.44 | 3.14 | 2.17 |
| 459 | 12.45 | 14.1 | 27.3 | 6.36 | 5.51 | 2.17 |

In Table 1, we compare the results numerically. Both the total worker time ratio and runtime ratio increase as $n$ grows. The total worker time ratio increases because GTA becomes less efficient, using more resources than necessary. The runtime ratio grows for ATA and ATA-Empirical since a larger

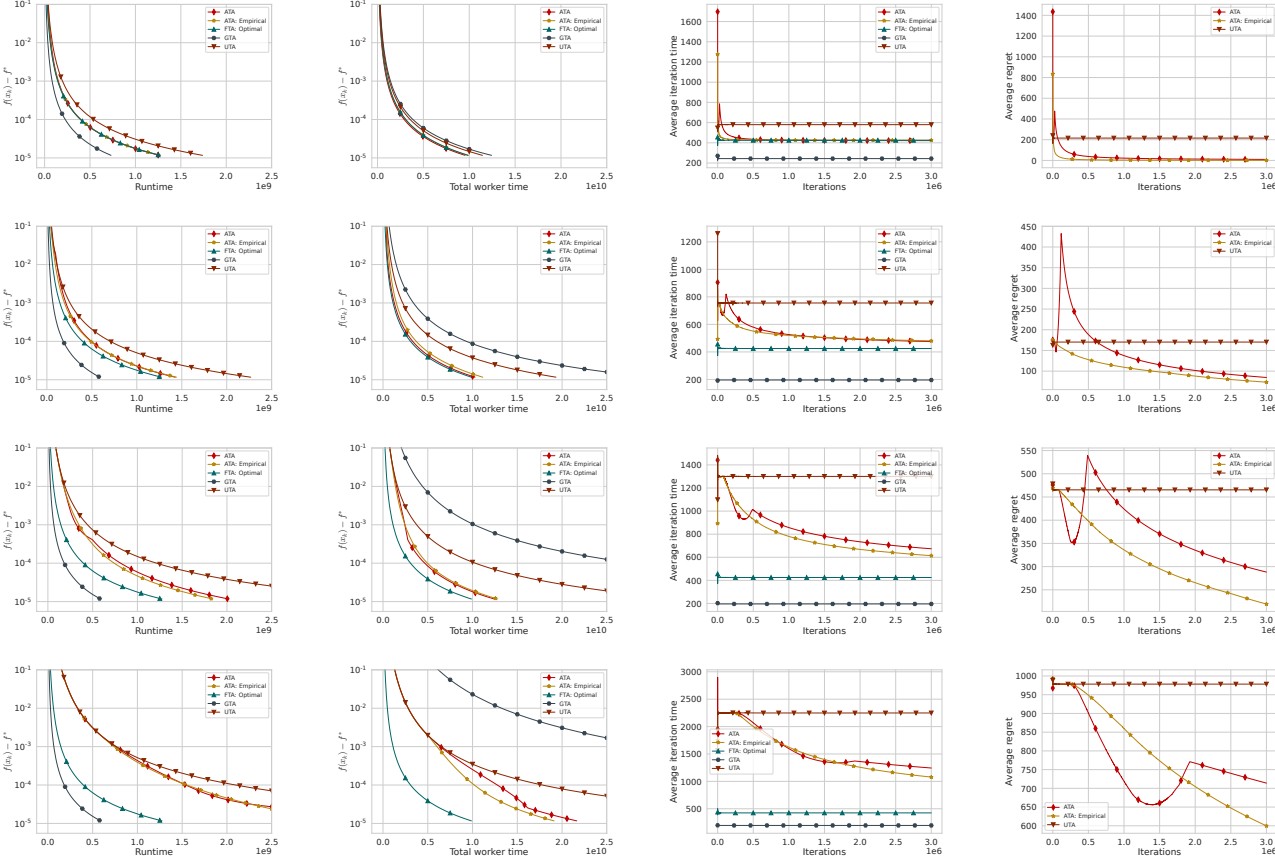

Figure 1: Each row increases the number of workers by a factor of 3, starting from 17, that is, $n = 17, 51, 153, 459$ from top to bottom. The first column shows runtime vs. suboptimality. The second column also plots suboptimality, but against total worker time, i.e., $\sum_{i=1}^{n} T_{i,k}$ in Algorithm 1. The third column presents the average iteration time, given by $C_k/k$ over all iterations $k$. The last column displays the averaged cumulative regret, as defined in (9).

number of workers requires more exploration. However, for OFTA this ratio remains unchanged, as discussed earlier.

We remark that in these experiments we started all runs for ATA and ATA-Empirical without prior knowledge of the computation time distribution. However, in real systems, where these algorithms are used multiple times, prior estimates of computation times from previous runs could be available. With this information, ATA and ATA-Empirical would be much faster, as they would spend less time on exploration, approaching the performance of OFTA in a faster way. We validate this through experiments presented in Appendix A.5.

In Appendix A.1, we conducted similar experiments with a different time distribution, where the mean times vary linearly across the arms. In Appendix A.2, we examine scenarios with varying client time distributions. Additionally, in Appendix A.3, we analyze regret performance, confirming its logarithmic behavior as predicted by Theorems 6.1 and 6.3. Finally, in Appendix A.4, we trained a simple CNN

on the CIFAR-100 dataset (Krizhevsky et al., 2009) using Adam (Kingma & Ba, 2014).

## Acknowledgments

The research reported in this publication was supported by funding from King Abdullah University of Science and Technology (KAUST): i) KAUST Baseline Research Scheme, ii) Center of Excellence for Generative AI, under award number 5940, iii) SDAIA-KAUST Center of Excellence in Artificial Intelligence and Data Science.

## Impact Statement

This paper presents work whose goal is to advance the field of Machine Learning. There are many potential societal consequences of our work, none which we feel must be specifically highlighted here.

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

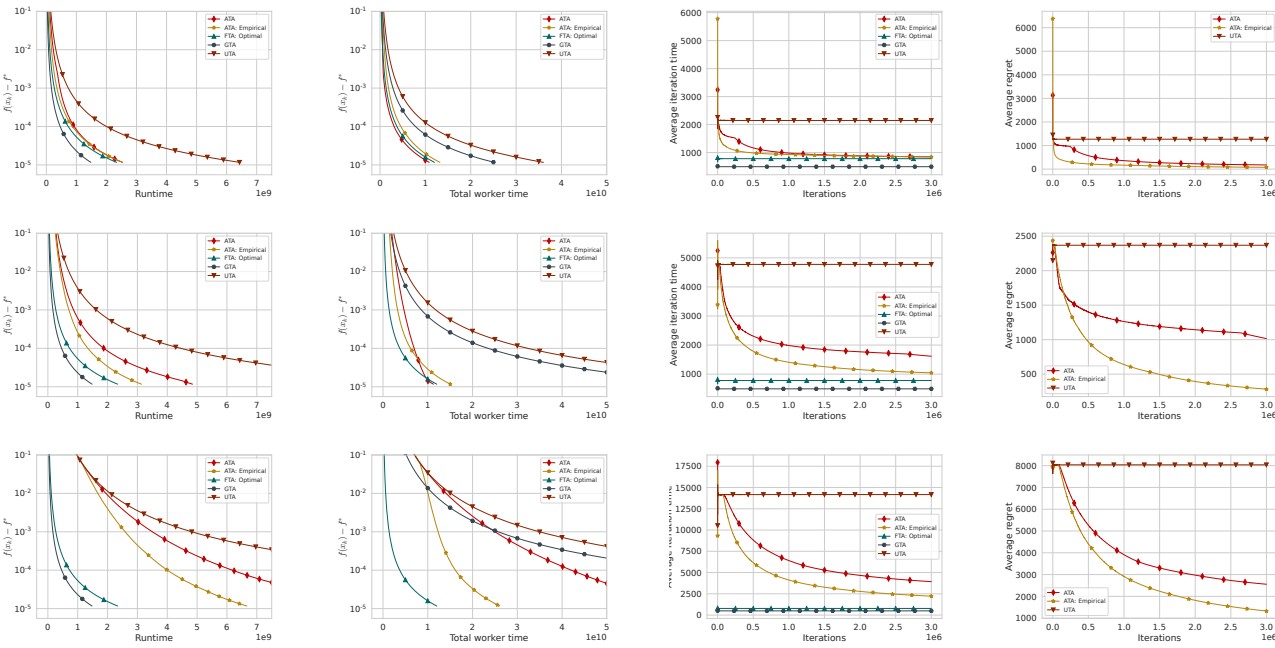

Figure 2: We use the same setup as in Figure 1, with each row tripling the number of workers, starting from $n = 17$.

## A. Additional Experiments

The objective function is a convex quadratic function $f : \mathbb{R}^d \to \mathbb{R}$ defined as

$$f(\boldsymbol{x}) = \frac{1}{2} x^\top \mathbf{A} x - \boldsymbol{b}^\top \boldsymbol{x},$$

where

$$\mathbf{A} = \frac{1}{4} \begin{bmatrix} 2 & -1 & & 0 \\ -1 & \ddots & \ddots & \\ & \ddots & \ddots & -1 \\ 0 & & -1 & 2 \end{bmatrix} \in \mathbb{R}^{d \times d}, \quad \text{and} \quad \boldsymbol{b} = \frac{1}{4} \begin{bmatrix} -1 \\ 0 \\ \vdots \\ 0 \end{bmatrix} \in \mathbb{R}^d .$$

We denote $f^*$ as the minimum value of the function $f$. Each of the $n$ workers is able to calculate unbiased stochastic gradients $\boldsymbol{g}(x)$ that satisfy

$$\mathbb{E}\left[ \|\boldsymbol{g}(x) - \nabla f(\boldsymbol{x})\|^2 \right] \leq 0.01^2 .$$

This is achieved by adding Gaussian noise to the gradients of $f$.

The experiments were implemented in Python. The distributed environment was emulated on machines with Intel(R) Xeon(R) Gold 6248 CPU @ 2.50GHz.

### A.1. Linear Noise

In this section we model the computation time for worker $i$ by the distribution

$$\nu_i = 29i + \text{Exp}(29i), \quad \text{for all} \quad i \in [n] .$$

The expected value of this distribution is $\mu_i = 2 \cdot 29i$ . Furthermore, the Orlicz norm satisfies the bound $\alpha_i \leq 2\mu_i$.

We again set $B = 23$ and run simulations similar to those in Section 7. The results are shown in Figure 2.

The important difference to the previous Figure 1 is that here ATA-Empirical performs better than ATA. This is because the Orlicz norm $\alpha = 4 \cdot 29n$ is much larger.

Similarly, we provide a numerical comparison in Table 2.

Table 2: This table presents ratios similar to those in Table 1.

| $n$ | TOTAL WORKER TIME RATIO | | | RUNTIME RATIO | | |
|---|---|---|---|---|---|---|
| | ATA | ATA-Empirical | OFTA | ATA | ATA-Empirical | OFTA |
| 17 | 2.32 | 1.91 | 2.1 | 1.71 | 1.71 | 1.58 |
| 51 | 6.71 | 5.02 | 6.29 | 3.27 | 2.12 | 1.58 |
| 153 | 3.41 | 8.68 | 18.87 | 7.96 | 4.5 | 1.58 |

## A.2. Heterogeneous Time Distributions

So far, we have only considered cases where clients follow the same distributions but with different means. In this section, we extend our experiments to cases where the distributions themselves differ. We consider five distributions: Exponential, Uniform, Half-Normal, Lognormal, and Gamma. We group five workers with these five distributions so that each group has the same mean, then vary the mean across different groups. More concretely, we use:

- $\mathrm{Exp}(c(5g + 1))$,

- $\mathrm{Uniform}\left(\frac{c(5g+1)}{2}, 3\frac{c(5g+1)}{2}\right)$,

- $\left|\mathcal{N}\left(0, c(5g + 1)\sqrt{\frac{\pi}{2}}\right)\right|$,

- $\mathrm{Lognormal}\left(\log(c(5g + 1))/2, \sqrt{\log(c(5g + 1))}\right)$,

- $\mathrm{Gamma}\left((c(5g + 1))^2, \frac{1}{c(5g+1)}\right)$ with shape and scale parameters.

Next, we add a constant $c(5g + 1)$ to all the distributions, where $c = 29$, and $g$ represents the group number, starting from 0. The clients are divided into $n/5$ groups.

The results of the experiments are shown in Figure 3. The plots demonstrate that the algorithms are robust across different distributions.

## A.3. Regret

In this section, we verify Theorems 6.1 and 6.3 on regret through simulations. We set $n = 20$ and $B = 5$, with the computation time for worker $i$ following the distribution

$$\nu_i = \mathrm{Exp}(2i), \quad \text{for all} \quad i \in [n].$$

We ran the simulation five times, and the plots include standard deviation bars, although they are not visible. The results are presented in Figure 4.

As expected, the regret grows logarithmically.

## A.4. Real Dataset

In this section, we present an experiment where we train a convolutional neural network (CNN) on the CIFAR-100 dataset (Krizhevsky et al., 2009). The network consists of three convolutional layers and two fully connected layers, with a total of 160k parameters.

We use the Adam optimizer (Kingma & Ba, 2014) with a constant step size of $8 \cdot 10^{-5}$. The computation time of the workers follows the same setup as in Figure 2. The results are shown in Figure 5.

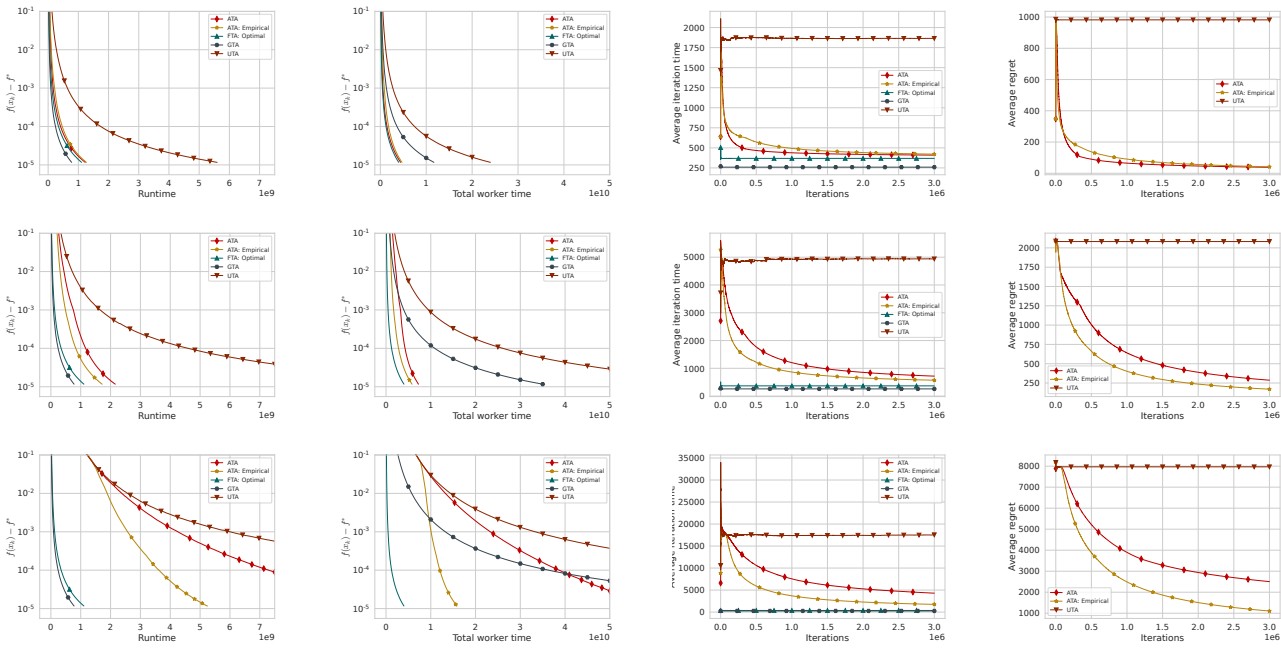

Figure 3: Each row corresponds to an increasing number of workers, with $n = 15, 45, 150$ from top to bottom. We consider five distributions—Exponential, Uniform, Half-Normal, Lognormal, and Gamma—grouping them to have the same mean and then varying the mean across different groups. The results demonstrate that the algorithms remain robust across different distributions. The columns represent the same as in Figure 1.

## A.5. Impact of Prior Knowledge on Time Distributions

In real-world systems where multiple machine learning models are trained, estimates of computation times from previous runs may be available. With this prior knowledge, ATA and ATA-Empirical can be much faster, as they spend less time on exploration and quickly approach the performance of OFTA.

To illustrate this, we vary the number of prior runs, $P$. Since our algorithms operate independently of the underlying optimization process, we first focus solely on the bandit component, updating the confidence scores of machines over several iterations. We then apply the loss curves to different segments of the bandit phase and compare the results as $P$ increases. A larger $P$ yields more accurate estimates.

The optimization setup remains the same as in Figure 5, with $B = 23$ and $n = 51$. The results are presented in Figure 6.

## B. Concrete Optimization Methods

In this section, we provide concrete examples of optimization algorithms using the ATA and GTA allocation strategies.

For optimization problems, we focus on SGD and Asynchronous SGD. Other methods, such as stochastic proximal point methods and higher-order methods, can be developed in a similar fashion.

### B.1. Stochastic Gradient Descent

For SGD, it is important to distinguish homogeneous and heterogeneous cases. Let us start from the homogeneous case.

#### B.1.1. HOMOGENEOUS REGIME

Consider the problem of finding an approximate stationary point of the optimization problem

$$\min_{\boldsymbol{x} \in \mathbb{R}^d} \left\{ f(\boldsymbol{x}) := \mathbb{E}_{\boldsymbol{\xi} \sim \mathcal{D}} \left[ f(\boldsymbol{x}; \boldsymbol{\xi}) \right] \right\}. \tag{12}$$

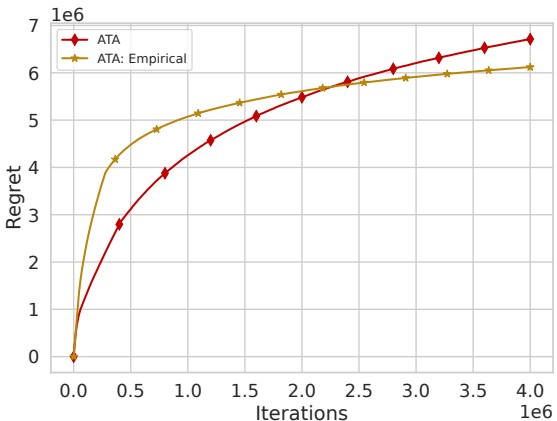

Figure 4: Regret growth over iterations.

We assume that each worker is able to compute stochastic gradient $f(\boldsymbol{x}; \boldsymbol{\xi})$ satisfying $\mathbb{E}_{\boldsymbol{\xi} \sim \mathcal{D}} \left[ \|f(\boldsymbol{x}; \boldsymbol{\xi}) - \nabla f(\boldsymbol{x})\|^2 \right] \leq \sigma^2$ for all $\boldsymbol{x} \in \mathbb{R}^d$.

In this case, SGD with allocation budget $B$ becomes Minibatch SGD with batch size $B$. The next step is determining how the batch is collected. For ATA, we refer to this method as SGD-ATA, as described in Algorithm 2.

---

**Algorithm 2** SGD-ATA (Homogeneous)

---

1: **Optimization inputs**: initial point $\boldsymbol{x}_0 \in \mathbb{R}^d$, stepsize $\gamma > 0$
2: **Allocation inputs**: allocation budget $B$
3: **Initialize**: empirical means $\hat{\mu}_{i,1} = 0$, usage counts $K_{i,1} = 0$, and usage times $T_{i,1} = 0$, for all $i \in [n]$
4: **for** $k = 1, \ldots, K$ **do**
5:     Compute LCBs $(s_{i,k})$ for all $i \in [n]$
6:     Find allocation: $\boldsymbol{a}_k \in \arg \min_{\boldsymbol{a} \in \mathcal{A}} \ell(\boldsymbol{a}, \boldsymbol{s}_k)$ .
7:     Allocate $a_{i,k}$ tasks to each worker $i \in [n]$
8:     Update $\boldsymbol{x}$:

$$\boldsymbol{x}_{k+1} = \boldsymbol{x}_k - \frac{\gamma}{B} \sum_{i=1}^{n} \sum_{j=1}^{a_{i,k}} \nabla f\left(\boldsymbol{x}_k; \boldsymbol{\xi}_i^j\right)$$

9:     **for** $i$ such that $a_{i,k} \neq 0$ **do**
10:         $K_{i,k+1} = K_{i,k} + a_{i,k}$
11:         $T_{i,k+1} = T_{i,k} + \sum_{j=1}^{a_{i,k}} X_{i,k}^{(j)}$
12:         $\hat{\mu}_{i,k+1} = \frac{T_{i,k+1}}{K_{i,k+1}}$
13:     **end for**
14: **end for**

---

In this case, each task consists in calculating the gradient using the device's local data, which is assumed to have the same distribution as the data on all other devices. Because of this, it does not matter which device performs the task. The method then averages these gradients to obtain an unbiased gradient estimator and performs a gradient descent step.

Now, let us give the version of Minibatch SGD using greedy allocation Algorithm 3.

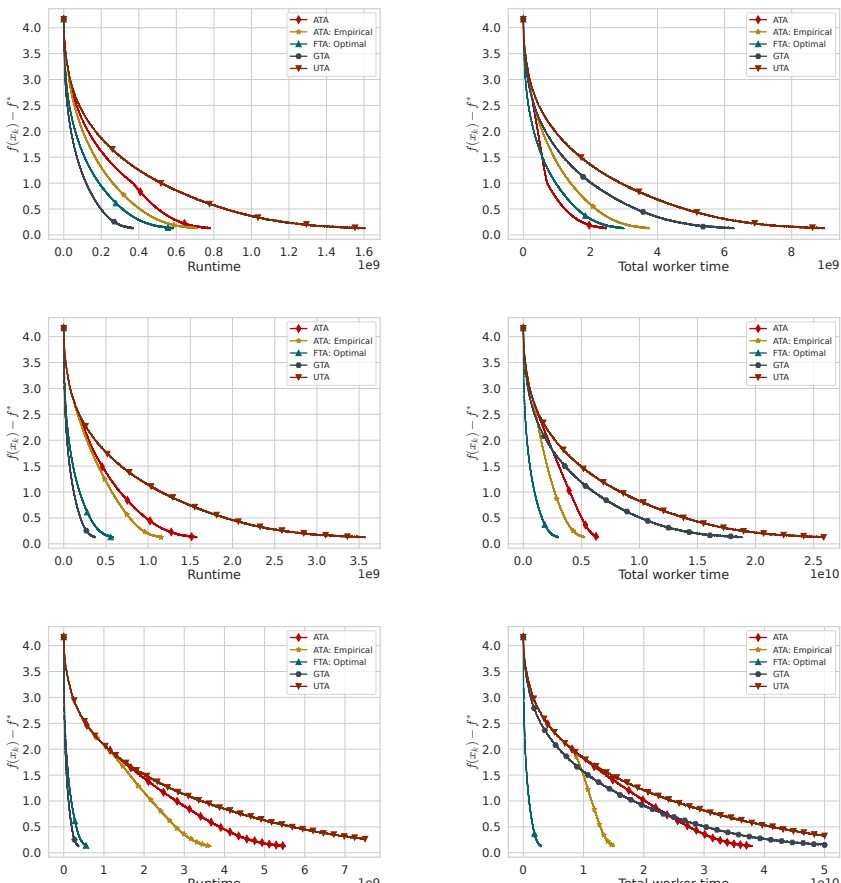

Figure 5: We use the CIFAR-100 dataset (Krizhevsky et al., 2009). The model is a CNN with three convolutional layers and two fully connected layers, totaling 160k parameters. The Adam optimizer (Kingma & Ba, 2014) is used with a constant step size of $8 \cdot 10^{-5}$. The computation time of the workers follows the same setup as in Figure 2, where the mean time increases linearly. The batch size remains the same at $B = 23$. Each row corresponds to a different number of workers, with $n = 17, 51, 153$ from top to bottom.

---

**Algorithm 3** SGD-GTA (Homogeneous)

---

1: **Input**: initial point $\boldsymbol{x}_0 \in \mathbb{R}^d$, stepsize $\gamma > 0$, allocation budget $B$
2: **for** $k = 1, \ldots, K$ **do**
3: $\quad b = 0$
4: $\quad$ Query single gradient from each worker $i \in [n]$
5: $\quad$ **while** $b < B$ **do**
6: $\quad\quad$ Gradient $\nabla f(\boldsymbol{x}_k; \boldsymbol{\xi}_{k_b})$ arrives from worker $i_{k_b}$
7: $\quad\quad \boldsymbol{g}_k = \boldsymbol{g}_k + \nabla f(\boldsymbol{x}_k; \boldsymbol{\xi}_{k_b}); \ \ b = b + 1$
8: $\quad\quad$ Query gradient at $\boldsymbol{x}_k$ from worker $i_{k_b}$
9: $\quad$ **end while**
10: $\quad$ Update the point: $\boldsymbol{x}_{k+1} = \boldsymbol{x}_k - \gamma \frac{\boldsymbol{g}^k}{B}$
11: **end for**

---

Algorithm 3 is exactly Rennala SGD method proposed by Tyurin & Richtárik (2024), which has optimal time complexity when the objective function is non-convex and smooth.

If the computation times are deterministic, then GTA makes the same allocation in each iteration. In that case, SGD-ATA will

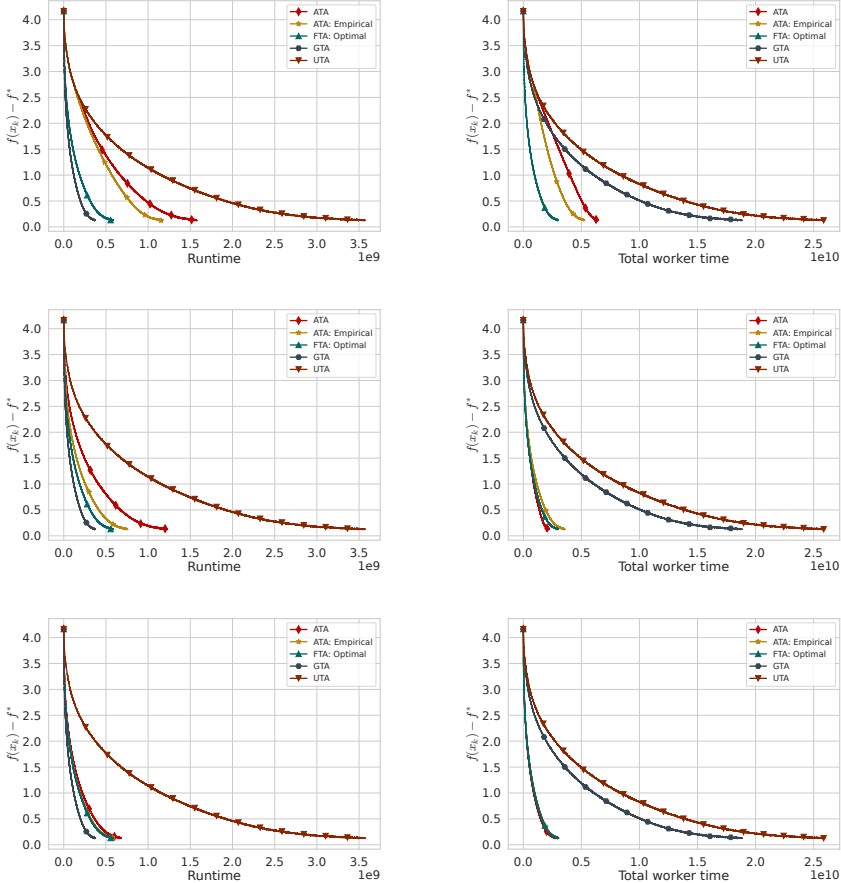

Figure 6: We use the same optimization setup as in Figure 5, with $B = 23$ and $n = 51$. The number of prior iterations, $P$, varies across rows, starting from the top with $P = 0, 5 \cdot 10^5, 5 \cdot 10^6$. As the number of prior iterations increases, we observe that the training of ATA and ATA-Empirical accelerates, bringing their performance closer to the optimal performance of OFTA.

converge to this fixed allocation. If the times are random, the allocation found by GTA may vary in each iteration. In this case, SGD-ATA will approach the best allocation for the expected times.

### B.1.2. HETEROGENEOUS REGIME

Now let us consider the following heterogeneous problem

$$\min_{x \in \mathbb{R}^d} \left\{ f(\boldsymbol{x}) := \frac{1}{n} \sum_{i=1}^{n} \mathbb{E}_{\boldsymbol{\xi}_i \sim \mathcal{D}_i} \left[ f_i(\boldsymbol{x}; \boldsymbol{\xi}_i) \right] \right\} .$$

Here each worker $i$ has its own data distribution $\mathcal{D}_i$.

We start with the greedy allocation. The algorithm is presented in Algorithm 4.

---

**Algorithm 4** SGD-GTA (Heterogeneous)

---

1: **Input**: initial point $\boldsymbol{x}_0 \in \mathbb{R}^d$, stepsize $\gamma > 0$, parameter $S$
2: **for** $k = 1, \ldots, K$ **do**
3:     $s_i = 0$ and $\boldsymbol{g}_{i,k} = \boldsymbol{0}$ for all $i \in [n]$
4:     Query single gradient from each worker $i \in [n]$
5:     **while** $\left( \frac{1}{n} \sum_{i=1}^{n} \frac{1}{s_i} \right)^{-1} < \frac{S}{n}$ **do**
6:         Gradient $\nabla f_j(\boldsymbol{x}_k; \boldsymbol{\xi}_k)$ arrives from worker $j$
7:         $\boldsymbol{g}_{j,k} = \boldsymbol{g}_{j,k} + \nabla f_j(\boldsymbol{x}_k; \boldsymbol{\xi}_k)$; $s_j = s_j + 1$
8:         Query gradient at $\boldsymbol{x}_k$ from worker $j$
9:     **end while**
10:     Update the point: $\boldsymbol{x}_{k+1} = \boldsymbol{x}_k - \gamma \frac{1}{n} \sum_{i=1}^{n} \frac{1}{s_i} \boldsymbol{g}_{i,k}$
11: **end for**

---

Algorithm 5 presents the Malenia SGD algorithm, proposed by Tyurin & Richtárik (2024), which is also optimal for non-convex smooth functions.

In each iteration, Algorithm 4 receives at least one gradient from each worker. Building on this idea, we design a method incorporating ATA, given in Algorithm 5.

---

**Algorithm 5** SGD-ATA (Heterogeneous)

---

1: **Optimization inputs**: initial point $\boldsymbol{x}_0 \in \mathbb{R}^d$, stepsize $\gamma > 0$
2: **Allocation inputs**: allocation budget $B$
3: **Initialize**: empirical means $\hat{\mu}_{i,1} = 0$, usage counts $K_{i,1} = 0$, and usage times $T_{i,1} = 0$, for all $i \in [n]$
4: **for** $k = 1, \ldots, K$ **do**
5:     Compute LCBs $(s_{i,k})$ for all $i \in [n]$
6:     Find allocation: $\boldsymbol{a}_k = \mathsf{RAS}(\boldsymbol{s}_k; B)$
7:     Allocate $a_{i,k} + 1$ tasks to each worker $i \in [n]$
8:     Update $\boldsymbol{x}$:

$$\boldsymbol{x}_{k+1} = \boldsymbol{x}_k - \frac{\gamma}{n} \sum_{i=1}^{n} \frac{1}{a_{i,k}+1} \sum_{j=1}^{a_{i,k}+1} \nabla f_i\left(\boldsymbol{x}_k; \boldsymbol{\xi}_i^j\right)$$

9:     For all $i \in [n]$, update:

$$K_{i,k+1} = K_{i,k} + a_{i,k}$$

$$T_{i,k+1} = T_{i,k} + \sum_{j=1}^{a_{i,k}} X_{i,k}^{(j)}$$

$$\hat{\mu}_{i,k+1} = \frac{T_{i,k+1}}{K_{i,k+1}}$$

10: **end for**

---

## B.2. Asynchronous SGD

Here, we focus on the homogeneous problem given in Equation (12). The greedy variant, Ringmaster ASGD, was proposed by Maranjyan et al. (2025b) and, like Rennala SGD, achieves the best runtime.

We now present its version with ATA, given in Algorithm 6.

Here, the task remains gradient computation, but each worker's subsequent tasks use different points for computing the gradient. These points depend on the actual computation times and the asynchronous nature of the method, hence the name

**Algorithm 6** ASGD-ATA

1: **Optimization inputs**: initial point $\boldsymbol{x}_0 \in \mathbb{R}^d$, stepsize $\gamma > 0$
2: **Allocation inputs**: allocation budget $B$
3: **Initialize**: empirical means $\hat{\mu}_{i,1} = 0$, usage counts $K_{i,1} = 0$, and usage times $T_{i,1} = 0$, for all $i \in [n]$
4: **for** $k = 1, \ldots, K$ **do**
5:     Compute LCBs $(s_{i,k})$ for all $i \in [n]$
6:     Find allocation: $\boldsymbol{a}_k = \mathsf{RAS}(\boldsymbol{s}_k; B)$
7:     Update $\boldsymbol{x}_k$ using Algorithm 7 with allocation $\boldsymbol{a}_k$
8:     For all $i$ such that $a_{i,k} \neq 0$, update:

$$K_{i,k+1} = K_{i,k} + a_{i,k}$$

$$T_{i,k+1} = T_{i,k} + \sum_{j=1}^{a_{i,k}} X_{i,k}^{(j)}$$

$$\hat{\mu}_{i,k+1} = \frac{T_{i,k+1}}{K_{i,k+1}}$$

9: **end for**

**Algorithm 7** ASGD

1: **Input:** Initial point $\boldsymbol{x}_0 \in \mathbb{R}^d$, stepsize $\gamma > 0$, allocation vector $\boldsymbol{a}$ with $\|\boldsymbol{a}\|_1 = B$
2: Workers with $a_i > 0$ start computing stochastic gradients at $\boldsymbol{x}_0$
3: **for** $s = 0, 1, \ldots, B - 1$ **do**
4:     Receive gradient $\nabla f(\boldsymbol{x}_{s+\delta_s}; \boldsymbol{\xi}_{s+\delta_s}^i)$ from worker $i$
5:     Update: $\boldsymbol{x}_{s+1} = \boldsymbol{x}_s - \gamma \nabla f(\boldsymbol{x}_{s+\delta_s}; \boldsymbol{\xi}_{s+\delta_s}^i)$
6:     **if** $a_i > 0$ **then**
7:         Worker $i$ begins computing $\nabla f(\boldsymbol{x}_{s+1}; \boldsymbol{\xi}_{s+1}^i)$
8:         Decrease remaining allocation for worker $i$ by one: $a_i = a_i - 1$
9:     **end if**
10: **end for**
11: **return:** $\boldsymbol{x}_B$

The sequence $\{\delta_s\}$ represents delays, where $\delta_s \geq 0$ is the difference between the iteration when worker $i$ started computing the gradient and iteration $s$, when it was applied.

Asynchronous SGD.

## C. Recursive Allocation Selection Algorithm

In this section, we introduce an efficient method for finding the best allocation. Given LCBs $\boldsymbol{s}_k$ and allocation budget $B$, each iteration of ATA (Algorithm 1) determines the allocation by solving

$$\boldsymbol{a}_k \in \arg\min_{\boldsymbol{a} \in \mathcal{A}} \ell(\boldsymbol{a}, \boldsymbol{s}_k),$$

where

$$\ell(\boldsymbol{a}, \boldsymbol{\mu}) := \max_{i \in [n]} a_i \mu_i = \|\boldsymbol{a} \odot \boldsymbol{\mu}\|_\infty,$$

with $\odot$ denoting the element-wise product. When clear from context, we write $\ell(\boldsymbol{a})$ instead of $\ell(\boldsymbol{a}, \boldsymbol{\mu})$.

In the early iterations, when some $s_i$ values are 0, ATA allocates uniformly across these arms until all $s_i$ values become positive. After that, the allocation is determined using the recursive routine in Algorithm 8.

*Remark* C.1. The iteration complexity of RAS is $\mathcal{O}(n \ln(\min\{B, n\}) + \min\{B, n\}^2)$. In fact, the first $n \ln(\min\{B, n\})$ term arises from identifying the smallest $B$ scores. For the second term, note that in (13), we have $r \leq \min\{B, n\}$.

### C.1. Optimality

We now prove that RAS finds the optimal allocation, as stated in the following lemma.

**Lemma C.2.** *For positive scores* $0 < s_1 \leq s_2 \leq \ldots \leq s_n$, RAS *(Algorithm 8) finds an optimal allocation* $\boldsymbol{h} \in \mathcal{A}$, *satisfying*

$$\boldsymbol{h} \in \arg\min_{\boldsymbol{a} \in \mathcal{A}} \|\boldsymbol{a} \odot \boldsymbol{s}\|_\infty .$$

*Proof.* We prove the claim by induction on the allocation budget $B$.

**Base Case** ($B = 1$)**:** When $B = 1$, RAS (Algorithm 8) allocates the task to worker with the smallest score (line 9). Thus, the base case holds.

**Inductive Step:** Assume that RAS finds an optimal allocation for budget $B - 1$, denoted by

$$\bar{\boldsymbol{h}} = \mathsf{RAS}(s_1, \ldots, s_n; B - 1) .$$

---

**Algorithm 8** Recursive Allocation Selection (RAS)

---

1: **Input:** Scores $s_1, \ldots, s_n$, allocation budget $B$
2: Assume without loss of generality that $s_1 \leq s_2 \leq \cdots \leq s_n$ (i.e., sort the scores)
3: **if** $B = 1$ **then**
4:     **return:** $(1, 0, \ldots, 0)$
5: **end if**
6: Find the previous best allocation:

$$\boldsymbol{a} = (a_1, \ldots, a_n) = \mathsf{RAS}\,(s_1, \ldots, s_n; B - 1)$$

7: Determine the first zero allocation:

$$r = \begin{cases} \min\{i \mid a_i = 0\}, & \text{if } a_n = 0 \\ n, & \text{otherwise} \end{cases} \tag{13}$$

8: Find the best next query allocation set:

$$M = \arg\min_{i \in [r]} \; \|(\boldsymbol{a} + \boldsymbol{e}_i) \odot \boldsymbol{s}\|_\infty\,,$$

    where $\boldsymbol{e}_i$ is the unit vector in direction $i$.
9: Select $j \in M$ such that the cardinality of

$$\arg\max_{i \in [r]} \; (a_i + e_{j,i})s_i$$

    is minimized
10: **return:** $\boldsymbol{a} + \boldsymbol{e}_j$

---

We need to prove that the solution returned for budget $B$, denoted by $\boldsymbol{h} = \bar{\boldsymbol{h}} + \boldsymbol{e}_r$, is also optimal.

Assume, for contradiction, that there exists $\boldsymbol{a} \in \mathcal{A}$ such that $\boldsymbol{a} \neq \boldsymbol{h}$ and $\ell(\boldsymbol{a}) < \ell(\boldsymbol{h})$. Write $\boldsymbol{a} = \bar{\boldsymbol{a}} + \boldsymbol{e}_q$ for some $q \in [n]$. Observe that $\|\bar{\boldsymbol{a}}\|_1 = B - 1$ because $\boldsymbol{a} \in \mathcal{A}$.

We consider two cases based on the value of $\ell\left(\bar{\boldsymbol{h}} + \boldsymbol{e}_r\right)$:

- $\ell\left(\bar{\boldsymbol{h}} + \boldsymbol{e}_r\right) = h_k s_k$ for some $k \neq r$. In this case, adding one unit to index $r$ does not change the maximum value, i.e., $\ell\left(\bar{\boldsymbol{h}}\right) = \ell\left(\bar{\boldsymbol{h}} + \boldsymbol{e}_r\right)$. By the inductive hypothesis, $\bar{\boldsymbol{h}}$ minimizes $\ell(\boldsymbol{x})$ for budget $B - 1$. Therefore, we have

$$\ell(\boldsymbol{a}) \geq \ell\left(\bar{\boldsymbol{a}}\right) \geq \ell\left(\bar{\boldsymbol{h}}\right) = \ell\left(\bar{\boldsymbol{h}} + \boldsymbol{e}_r\right) = \ell(\boldsymbol{h}),$$

    which contradicts the assumption that $\ell(\boldsymbol{a}) < \ell(\boldsymbol{h})$.

- $\ell\left(\bar{\boldsymbol{h}} + \boldsymbol{e}_r\right) = \left(\bar{h}_r + 1\right) s_r$. By the algorithm's logic, $\left(\bar{h}_r + 1\right) s_r \leq \left(\bar{h}_i + 1\right) s_i$ for all $i \neq r$. Since $\ell(\bar{\boldsymbol{h}} + \boldsymbol{e}_r) \leq \ell(\bar{\boldsymbol{h}} + \boldsymbol{e}_q)$ and we assumed $\ell(\bar{\boldsymbol{a}} + \boldsymbol{e}_q) = \ell(\boldsymbol{a}) < \ell(\boldsymbol{h}) = \ell(\bar{\boldsymbol{h}} + \boldsymbol{e}_r)$, then $\bar{\boldsymbol{a}} \neq \bar{\boldsymbol{h}}$ otherwise $\ell(\bar{\boldsymbol{a}} + \boldsymbol{e}_q) < \ell(\bar{\boldsymbol{a}} + \boldsymbol{e}_r)$. Given that $\|\bar{\boldsymbol{h}}\|_1 = \|\bar{\boldsymbol{a}}\|_1$, this implies that there exists some $u \in [n]$ such that $0 \leq \bar{a}_u \leq \bar{h}_u - 1$ and another index $v \in [n]$ where $\bar{a}_v \geq \bar{h}_v + 1$.

    In addition, note that $r$ is chosen such that $\ell\left(\bar{\boldsymbol{h}} + \boldsymbol{e}_r\right)$ is minimum. Using the fact that $\ell\left(\bar{\boldsymbol{h}} + \boldsymbol{e}_r\right) = \left(\bar{h}_r + 1\right) s_r$, we have that for any index $q$, we also necessarily have $\ell\left(\bar{\boldsymbol{h}} + \boldsymbol{e}_q\right) = \left(\bar{h}_q + 1\right) s_q$. Using this, we deduce

$$\ell(\boldsymbol{h}) = \ell\left(\bar{\boldsymbol{h}} + \boldsymbol{e}_r\right) \leq \ell\left(\bar{\boldsymbol{h}} + \boldsymbol{e}_v\right) = \left(\bar{h}_v + 1\right) s_v \leq \max_i \; \bar{a}_i s_i = \ell\left(\bar{\boldsymbol{a}}\right) \leq \ell(\boldsymbol{a}),$$

    where in the second inequality we used the fact that $\bar{a}_v \geq \bar{h}_v + 1$ and in the last inequality we used the fact that the loss is not decreasing for we add one element to the vector. This chain of inequalities again contradicts the assumption that $\ell(\boldsymbol{a}) < \ell(\boldsymbol{h})$.

Since both cases lead to contradictions, we conclude that no $\boldsymbol{a} \in \mathcal{A}$ exists with $\ell(\boldsymbol{a}) < \ell(\boldsymbol{h})$. Thus, RAS produces an optimal allocation for budget $B$. $\qquad\square$

## C.2. Minimal Cardinality

Among all possible allocations RAS choose one that always minimizes the cardinality of the set:

$$\arg\max_{i\in[n]} a_i s_i .$$

The reason for this choice is just technical as it allows the Lemma D.1 to be true.

**Lemma C.3.** *The output of RAS ensures the smallest cardinality of the set:*

$$\arg\max_{i\in[n]} a_i s_i$$

*among all the optimal allocations $\boldsymbol{a}$.*

*Proof.* This proof uses similar reasoning as the one before.

Let $\boldsymbol{h} = \mathsf{RAS}(\boldsymbol{s}; B)$, and denote the cardinality of the set $\arg\max_{i\in[n]} a_i s_i$ for allocation $\boldsymbol{a}$ by

$$C_B(\boldsymbol{a}) = \left| \arg\max_{i\in[n]} a_i s_i \right| \geq 1 .$$

We prove the claim by induction on $B$.

**Base Case ($B = 1$):** For $B = 1$, there is a single coordinate allocation, thus $C_1(\boldsymbol{h}) = 1$, which is the smallest possible cardinality.

**Inductive Step:** Assume that RAS finds an optimal allocation for budget $B - 1$ with the smallest cardinality, denote its output by

$$\bar{\boldsymbol{h}} = \mathsf{RAS}(s_1, \ldots, s_n; B - 1) .$$

We need to prove that $\boldsymbol{h} = \bar{\boldsymbol{h}} + \boldsymbol{e}_r$ minimizes $C_B(\boldsymbol{a})$ among all optimal allocations for budget $B$.

Assume, for contradiction, that there exists $\boldsymbol{a} \in \mathcal{A}$ such that $\boldsymbol{a} \neq \boldsymbol{h}$, $\ell(\boldsymbol{a}) = \ell(\boldsymbol{h})$, and $C_B(\boldsymbol{a}) < C_B(\boldsymbol{h})$. Write $\boldsymbol{a} = \bar{\boldsymbol{a}} + \boldsymbol{e}_q$ for some $q \in [n]$. We consider three cases:

- $C_B(\boldsymbol{h}) = 1$. Since the minimum cardinality is exactly 1, we must have $C_B(\boldsymbol{a}) \geq 1 = C_B(\boldsymbol{h})$, that contradicts our assumption.

- $C_B(\boldsymbol{h}) = C_{B-1}(\bar{\boldsymbol{h}}) > 1$. This occurs when $\ell(\boldsymbol{h}) = \ell(\bar{\boldsymbol{h}}) \neq (\bar{h}_r + 1) s_r$. By the optimality of $\boldsymbol{h}$, we have $\ell(\bar{\boldsymbol{h}}) \leq \ell(\bar{\boldsymbol{a}}) \leq \ell(\boldsymbol{a}) = \ell(\boldsymbol{h}) = \ell(\bar{\boldsymbol{h}})$, which implies $\ell(\bar{\boldsymbol{a}}) = \ell(\boldsymbol{a})$. Therefore, $C_{B-1}(\bar{\boldsymbol{a}}) \leq C_B(\boldsymbol{a})$. Since the induction hypothesis holds for $B - 1$, we have $C_{B-1}(\bar{\boldsymbol{h}}) \leq C_{B-1}(\bar{\boldsymbol{a}})$. Thus,

$$C_B(\boldsymbol{h}) = C_{B-1}(\bar{\boldsymbol{h}}) \leq C_{B-1}(\bar{\boldsymbol{a}}) \leq C_B(\boldsymbol{a}),$$

  which leads to a contradiction.

- $C_B(\boldsymbol{h}) = C_{B-1}(\bar{\boldsymbol{h}}) + 1$. This occurs when $\ell(\boldsymbol{h}) = \ell(\bar{\boldsymbol{h}}) = (\bar{h}_r + 1) s_r$. Proceeding as in the previous case, we have $\ell(\bar{\boldsymbol{a}}) = \ell(\boldsymbol{a})$, and hence $C_{B-1}(\bar{\boldsymbol{a}}) \leq C_B(\boldsymbol{a})$. Since the induction hypothesis holds for $B - 1$, we know $C_{B-1}(\bar{\boldsymbol{h}}) \leq C_{B-1}(\bar{\boldsymbol{a}})$.

  We now have additional cases:

  - If $C_{B-1}(\bar{\boldsymbol{a}}) = C_{B-1}(\bar{\boldsymbol{h}}) + 1$, then

  $$C_B(\boldsymbol{h}) = C_{B-1}(\bar{\boldsymbol{h}}) + 1 = C_{B-1}(\bar{\boldsymbol{a}}) \leq C_B(\boldsymbol{a}),$$

    which leads to a contradiction.

– Now assume $C_{B-1}(\bar{\boldsymbol{a}}) = C_{B-1}(\bar{\boldsymbol{h}})$. We will show that in this case, $C_B(\boldsymbol{a}) = C_{B-1}(\bar{\boldsymbol{a}}) + 1$. By contradiction, suppose $C_B(\boldsymbol{a}) = C_{B-1}(\bar{\boldsymbol{a}})$, which implies $(\bar{a}_q + 1)s_q < \ell(\boldsymbol{a})$. Let $k$ be an index such that $\bar{a}_k s_k = \ell(\boldsymbol{a})$. Construct a new allocation $\boldsymbol{a}' = \bar{\boldsymbol{a}} + \boldsymbol{e}_q - \boldsymbol{e}_k$. Then,

$$C_{B-1}(\boldsymbol{a}') = C_{B-1}(\bar{\boldsymbol{a}}) - 1 < C_{B-1}(\bar{\boldsymbol{h}}),$$

which contradicts the induction hypothesis. Thus, $C_B(\boldsymbol{a}) = C_{B-1}(\bar{\boldsymbol{a}}) + 1$. Using this, we have

$$C_B(\boldsymbol{h}) = C_{B-1}(\bar{\boldsymbol{h}}) + 1 = C_{B-1}(\bar{\boldsymbol{a}}) + 1 = C_B(\boldsymbol{a}),$$

which again contradicts $C_B(\boldsymbol{a}) < C_B(\boldsymbol{h})$.

This concludes the proof. $\qquad\square$

## D. Proofs of Theorem 6.1, Theorem 6.3, and Theorem 4.2

We start by recalling the notation. For $i \in [n]$ and $k \in [K]$, $(X_{i,k}^{(u)})_{u \in [B]}$ denote $B$ independent samples at round $k$ from distribution $\nu_i$. When using an allocation vector $\boldsymbol{a}_k \in \mathcal{A}$, the total computation time of worker $i$ at round $k$ is $\sum_{u=1}^{a_{i,k}} X_{i,k}^{(u)}$, when $a_{i,k} > 0$. $\boldsymbol{\mu} = (\mu_1, \ldots, \mu_K)$ is the vector of means. For each $k \in [K]$, when using the allocation vector $\boldsymbol{a}_k$, we recall the definition of the proxy loss $\ell : \mathcal{A} \times \mathbb{R}_{\geq 0}^n \to \mathbb{R}_{\geq 0}$ by

$$\ell(\boldsymbol{a}_k, \boldsymbol{\lambda}) = \max_{i \in [n]} a_{i,k} \lambda_i,$$

where $\boldsymbol{\lambda} = (\lambda_1, \ldots, \lambda_n)$ is a vector of non-negative components. When $\boldsymbol{\lambda} = \boldsymbol{\mu}$, we drop the dependence on the second input of $\ell$. For each $\boldsymbol{\lambda}$, let $\bar{\boldsymbol{a}}_{\boldsymbol{\lambda}} \in \mathcal{A}$, be the action minimizing this loss

$$\bar{\boldsymbol{a}}_{\boldsymbol{\lambda}} \in \operatorname*{arg\,min}_{\boldsymbol{a} \in \mathcal{A}} \ell(\boldsymbol{a}, \boldsymbol{\lambda}).$$

We drop the dependency on $\boldsymbol{\mu}$ from $\bar{\boldsymbol{a}}_{\boldsymbol{\mu}}$ to ease notation. The actual (random) computation time at round $k$ is denoted by $C : \mathcal{A} \to \mathbb{R}_+$:

$$C(\boldsymbol{a}_k) := \max_{i \in [n]} \sum_{u=1}^{a_{i,k}} X_{i,k}^{(u)}. \tag{14}$$

Let $\boldsymbol{a}^*$ be the action minimizing the expected time

$$\boldsymbol{a}^* \in \operatorname*{arg\,min}_{\boldsymbol{a} \in \mathcal{A}} \mathbb{E}[C(\boldsymbol{a})].$$

The expected regret after $K$ rounds is defined as follows

$$\mathcal{R}_K := \sum_{t=1}^{K} \mathbb{E}[\ell(\boldsymbol{a}_k) - \ell(\bar{\boldsymbol{a}})].$$

For the remainder of this analysis we consider $\bar{\boldsymbol{a}} \in \operatorname*{arg\,min}_{\boldsymbol{a} \in \mathcal{A}} \ell(\boldsymbol{a})$ found using the RAS procedure. For each $i \in [n]$, recall that $k_i$ is the smallest integer such that

$$(\bar{a}_i + k_i)\mu_i > \ell(\bar{\boldsymbol{a}}). \tag{15}$$

Below we present a technical lemma used in the proofs of Theorems 6.1 and 6.3.

**Lemma D.1.** *Let $\boldsymbol{x} = (x_1, \ldots, x_n) \in \mathbb{R}_{\geq 0}^n$. Let $\boldsymbol{a}$ be the output of* RAS$(\boldsymbol{x}; B)$. *For each $i, j \in [n]$, we have*

$$a_j x_j \leq (a_i + 1)x_i.$$

*Proof.* Fix $\boldsymbol{x} \in \mathbb{R}_+^n$, and let $\boldsymbol{a} = $ RAS$(\boldsymbol{x}; B)$. The result is straightforward when $\min_{i \in [n]} x_i = 0$.

Suppose that $x_i > 0$ for all $i \in [n]$. Let $s \geq 1$ denote the cardinality

$$s := \left| \arg\max_{i \in [n]} a_i x_i \right| .$$

Fix $i, j \in [n]$, let $k \in \arg\max_{i \in [n]} a_i x_i$. We need to show that

$$a_k x_k \leq (a_i + 1) x_i .$$

We use a proof by contradiction. Suppose that we have $a_k x_k > (a_i + 1) x_i$ consider the allocation vector $\boldsymbol{a}' \in \mathcal{A}$ given by $a_k' = a_k - 1$, $a_i' = a_i + 1$ and $a_u' = \bar{a}_u$ when $u \notin \{i, k\}$. Let $R := \max_{u \neq i,k} \{a_u x_u\}$. We have

$$\ell(\boldsymbol{a}', \boldsymbol{x}) = \max_{u \in [n]} a_u' x_u = \max\{(a_i + 1) x_i, (a_k - 1) x_k, R\} .$$

We consider two cases:

- Suppose that $s = 1$ (i.e., the only element in $[n]$ such that $a_u x_u = \ell(\boldsymbol{a}, \boldsymbol{x})$ is $k$), then we have necessarily $R < a_k x_k$. Moreover, by the contradiction hypothesis, $(a_i + 1) x_i < a_k x_k$. Therefore,

$$\ell(\boldsymbol{a}', \boldsymbol{x}) = \max\{(a_i + 1) x_i, (a_k - 1) x_k, R\} < a_k x_k = \ell(\boldsymbol{a}, \boldsymbol{x}),$$

  which contradicts the definition of $\boldsymbol{a}$.

- Suppose that $s \geq 2$, since by hypothesis $a_k x_k > (a_i + 1) x_i$, we clearly have $a_i x_i < \ell(\boldsymbol{a}, \boldsymbol{x})$ therefore among the set $[n] \setminus \{k, i\}$ there are exactly $s - 1$ elements such that $a_u x_u = \ell(\boldsymbol{a}, \boldsymbol{x})$. In particular, this gives

$$\ell(\boldsymbol{a}', \boldsymbol{x}) = \max_{u \in [n]} \{(a_i + 1) x_i, (a_k - 1) x_k, R\} = R = \ell(\boldsymbol{a}, \boldsymbol{x}) .$$

  Therefore, $\boldsymbol{a}' \in \arg\min_{\boldsymbol{a} \in \mathcal{A}} \ell(\boldsymbol{a}, \boldsymbol{x})$ and the number of elements such that $a_i' x_i = \ell(\boldsymbol{a}', \boldsymbol{x}) = \ell(\boldsymbol{a}, \boldsymbol{x})$ is at most $s - 1$, which contradicts the fact that $s$ is minimal given the RAS choice and Lemma C.3.

As a conclusion we have $a_k x_k \leq (a_i + 1) x_i$. $\qquad \square$

*Remark D.2.* Recall that Lemma D.1 guarantees that $k_i$ defined in (15) satisfy: $k_i \in \{1, 2\}$ for each $i \in [n]$.

### D.1. Proof of Theorem 6.1

Below we restate the theorem.

**Theorem 6.1.** *Suppose that Assumption 3.1 holds. Let $\bar{\boldsymbol{a}} \in \arg\min_{\boldsymbol{a} \in \mathcal{A}} \ell(\boldsymbol{a})$, in case of multiple optimal actions, we consider the one output by RAS when fed with $\boldsymbol{\mu}$. Then, the expected regret of ATA with inputs $(B, \alpha)$ satisfies*

$$\mathcal{R}_K \leq 2n \max_{i \in [n]} \{B\mu_i - \ell(\bar{\boldsymbol{a}})\} + c \cdot \sum_{i=1}^{n} \frac{\alpha^2 (\bar{a}_i + k_i)(B\mu_i - \ell(\bar{\boldsymbol{a}}))}{((\bar{a}_i + k_i)\mu_i - \ell(\bar{\boldsymbol{a}}))^2} \cdot \ln K,$$

*where $\alpha = \max_{i \in [n]} \|X_i - \mu_i\|_{\psi_1}$, and $c$ is a constant.*

*Proof.* Let $K_{i,k}$ be the number of rounds where arm $i$ was queried prior to round $k$ (we take $K_{i,1} = 0$). Recall that we chose the following confidence bound: if $K_{i,k} \geq 1$, then

$$\mathrm{conf}(i, k) = 2\alpha \sqrt{\frac{\ln(2k^2)}{K_{i,k}}} + 2\alpha \frac{\ln(2k^2)}{K_{i,k}},$$

and $\mathrm{conf}(i, k) = \infty$ otherwise. Recall that $\hat{\mu}_{i,k}$ denotes the empirical mean of samples from $\nu_i$ observed prior to $k$ if $K_{i,k} \geq 0$ and $\hat{\mu}_{i,k} = 0$ if $K_{i,k} = 0$. Let $s_{i,k}$ denote the lower confidence bound used in the algorithm:

$$s_{i,k} = (\hat{\mu}_{i,k} - \mathrm{conf}(i, k))_+ .$$

We introduce the events $\mathcal{E}_{i,k}$ for $i \in [n]$ and $k \in [K]$ defined by

$$\mathcal{E}_{i,k} := \{|\hat{\mu}_{i,k} - \mu_i| > \mathrm{conf}(i,k)\} .$$

Let

$$\mathcal{E}_k = \cup_{i \in [n]} \mathcal{E}_{i,k}.$$

Let us prove that for each $k \in [K]$ and $i \in [n]$: $\mathbb{P}(\mathcal{E}_{i,k}) \leq \frac{1}{k^2}$, which gives using a union bound $\mathbb{P}(\mathcal{E}_k) \leq \frac{n}{k^2}$. Let $i \in [n]$, using Lemma E.1 and taking $\delta = 1/k^2$, we have

$$\mathbb{P}(\mathcal{E}_{i,k}) = \mathbb{P}\{|\hat{\mu}_{i,k} - \mu| > \mathrm{conf}(i,k)\} \leq \frac{1}{k^2} .$$

We call a "bad round", a round $k$ where we have $\ell(\boldsymbol{a}_k) > \ell(\bar{\boldsymbol{a}})$. Let us upper bound the number of bad rounds.

Observe that in a bad round there is necessarily an arm $i \in [K]$ such that $a_{i,k}\mu_i > \ell(\bar{\boldsymbol{a}})$. For each $i \in [n]$, let $N_i(k)$ denote the number of rounds $q \in \{1, \ldots, k\}$ where $a_{i,q}\mu_i > \ell(\bar{\boldsymbol{a}})$ and $i \in \arg\max_{j \in [n]} a_{j,q}\mu_j$ (this corresponds to a bad round triggered by worker $i$)

$$N_i(k) := |\{q \in \{1, \ldots, k\} : a_{i,q}\mu_i > \ell(\bar{\boldsymbol{a}}) \text{ and } a_{i,q}\mu_i = \ell(\boldsymbol{a}_q)\}| .$$

We show that in the case of $\ell(\boldsymbol{a}_k) > \ell(\bar{\boldsymbol{a}})$, the following event will hold: there exists $i \in [n]$ such that

$$E_{i,k} := \mathcal{E}_k \text{ or } \left\{ N_i(k-1) \leq \frac{24\alpha^2(\bar{a}_i + k_i)\ln(2K^2)}{((\bar{a}_i + k_i)\mu_i - \ell(\bar{\boldsymbol{a}}))^2} \right\} .$$

To prove this we use a contradiction argument. Suppose that for each $i \in [n]$, $\neg E_{i,k}$ holds and that $\ell(\boldsymbol{a}_k) > \ell(\bar{\boldsymbol{a}})$. This means that $k$ is a bad round, let $i$ be an arm that triggered this bad round (i.e., $i \in \arg\max_{j \in [n]} a_{j,k}\mu_j$). Event $\neg E_{i,k}$ gives in particular

$$N_i(k-1) > \frac{24\alpha^2(\bar{a}_i + k_i)\ln(2K^2)}{((\bar{a}_i + k_i)\mu_i - \ell(\bar{\boldsymbol{a}}))^2} . \tag{16}$$

Observe that in each round where $N_i(\cdot)$ is incremented, the number of samples received from the distribution $\nu_i$ increases by at least $\bar{a}_i + k_i$. Therefore, we have (16) implies

$$K_{i,k} > \frac{24\alpha^2(\bar{a}_i + k_i)^2 \ln(2K^2)}{((\bar{a}_i + k_i)\mu_i - \ell(\bar{\boldsymbol{a}}))^2} = \frac{24\alpha^2 \ln(2K^2)}{\left(\mu_i - \frac{\ell(\bar{\boldsymbol{a}})}{\bar{a}_i + k_i}\right)^2} .$$

Then we have, using the expressions of $\mathrm{conf}(\cdot)$ and the bound above

$$2\mathrm{conf}(i,k) = 4\alpha\sqrt{\frac{\ln(2k^2)}{K_{i,k}}} + 4\alpha\frac{\ln(2k^2)}{K_{i,k}}$$

$$\leq \mu_i - \frac{\ell(\bar{\boldsymbol{a}})}{\bar{a}_i + k_i} . \tag{17}$$

The contradiction hypothesis gives that $a_{i,k}\mu_i > \ell(\bar{\boldsymbol{a}})$, then we have, using the definition of $k_i$ that $a_{i,k} \geq \bar{a}_i + k_i$. Therefore, (17) gives

$$2\mathrm{conf}(i,k) < \mu_i - \frac{\ell(\bar{\boldsymbol{a}})}{a_{i,k}} . \tag{18}$$

Observe that in each round $\|\boldsymbol{a}_k\|_0 = B$, therefore if we have $a_{i,k} \geq \bar{a}_i + k_i > \bar{a}_i$ for some $i$, we necessarily have that there exists $j \in [n] \setminus \{i\}$ such that $a_{j,k} \leq \bar{a}_j - 1$. Using the fact that $\ell(\bar{\boldsymbol{a}}) \geq \bar{a}_j\mu_j$ with (18), we get

$$a_{i,k}(\mu_i - 2\mathrm{conf}(i,k)) > \bar{a}_j\mu_j . \tag{19}$$

Since both $\neg\mathcal{E}_{i,k}$ and $\neg\mathcal{E}_{j,k}$ hold (because $\neg E_{i,k}$ implies $\neg\mathcal{E}_k$), we have that

$$\mu_i - 2\mathrm{conf}(i,k) \leq \hat{\mu}_{i,k} - \mathrm{conf}(i,k) \leq s_{i,k}, \tag{20}$$

and $\mu_j \geq \hat{\mu}_{j,k} - \text{conf}(j, k)$. Recall that $\mu_j \geq 0$, therefore

$$\mu_j \geq (\hat{\mu}_{j,k} - \text{conf}(j, k))_+ = s_{j,k} . \tag{21}$$

Using the bounds (20) and (21) in (19), we have

$$a_{i,k}s_{i,k} > \bar{a}_j s_{j,k} \geq (a_{j,k} + 1)s_{j,k},$$

where we used the definition of $j$ in the second inequality. This contradicts the statement of Lemma D.1, which concludes the contradiction argument. Therefore, the event that $k$ is a bad round implies that $E_{i,k}$ holds for at least one $i \in [n]$. We say that a bad round was triggered by arm $i$, a round where $N_i(\cdot)$ was incremented. Observe that if $k \in [K]$ is not a bad round then $\mathbb{E}\left[\ell(\boldsymbol{a}_k)\right] - \ell(\bar{\boldsymbol{a}}) = 0$, otherwise if $k$ is a bad round triggered by $i \in [n]$ then $\mathbb{E}\left[\ell(\boldsymbol{a}_k)\right] - \ell(\bar{\boldsymbol{a}}) \leq B\mu_i - \ell(\bar{\boldsymbol{a}})$. To ease notation we introduce for $i \in [n]$

$$H_i := \frac{24\alpha^2(\bar{a}_i + k_i)\ln(2K^2)}{\left((\bar{a}_i + k_i)\mu_i - \ell(\bar{\boldsymbol{a}})\right)^2} .$$

The expected regret satisfies

$$
\begin{aligned}
\mathcal{R}_K &= \sum_{i=1}^{K} \mathbb{E}\left[\ell(\boldsymbol{a}_k) - \ell(\bar{\boldsymbol{a}})\right] \\
&\leq \sum_{i=1}^{n} (B\mu_i - \ell(\bar{\boldsymbol{a}}))\mathbb{E}[N_i(K)] \\
&= \sum_{i=1}^{n}\sum_{k=1}^{K} (B\mu_i - \ell(\bar{\boldsymbol{a}}))\mathbb{E}\left[\mathbb{1}(k \text{ is a bad round triggered by } i)\right] \\
&\leq \max_{i \in [n]}\{(B\mu_i - \ell(\bar{\boldsymbol{a}}))\} \cdot \sum_{t=1}^{K}\mathbb{P}(\mathcal{E}_k) + \sum_{i=1}^{n}(B\mu_i - \ell(\bar{\boldsymbol{a}}))\sum_{k=1}^{K}\mathbb{E}\left[\mathbb{1}(k \text{ is a bad round triggered by } i) \mid \neg\mathcal{E}_k\right] \\
&\leq \max_{i \in [n]}\{(B\mu_i - \ell(\bar{\boldsymbol{a}}))\} \cdot \sum_{t=1}^{K}\mathbb{P}(\mathcal{E}_k) + \sum_{i=1}^{n}(B\mu_i - \ell(\bar{\boldsymbol{a}}))\sum_{k=1}^{K}\mathbb{E}\left[\mathbb{1}(N_i(k) = 1 + N_i(k-1) \text{ and } N_i \leq H_i) \mid \neg\mathcal{E}_k\right] \\
&\leq \max_{i \in [n]}\{(B\mu_i - \ell(\bar{\boldsymbol{a}}))\} \cdot \sum_{k=1}^{K}\mathbb{P}(\mathcal{E}_k) + \sum_{i=1}^{n}(B\mu_i - \ell(\bar{\boldsymbol{a}}))H_i \\
&\leq 2n \max_{i \in [n]}\{(B\mu_i - \ell(\bar{\boldsymbol{a}}))\} + \sum_{i=1}^{n}\frac{24\alpha^2(\bar{a}_i + k_i)(B\mu_i - \ell(\bar{\boldsymbol{a}}))\ln(2K^2)}{\left((\bar{a}_i + k_i)\mu_i - \ell(\bar{\boldsymbol{a}})\right)^2} . \qquad \square
\end{aligned}
$$

### D.2. Proof of Theorem 6.3

**Theorem 6.3.** *Suppose that Assumption 3.1 holds. Let $\bar{\boldsymbol{a}} \in \arg\min_{\boldsymbol{a} \in \mathcal{A}} \ell(\boldsymbol{a})$, in case of multiple optimal actions, we consider the one output by* RAS *when fed with* $\boldsymbol{\mu}$. *Then, the expected regret of* ATA-Empirical *with the empirical confidence bounds using the inputs* $(B, \eta)$ *satisfies*

$$\mathcal{R}_K \leq 2n \max_{i \in [n]}\{B\mu_i - \ell(\bar{\boldsymbol{a}})\} + c \cdot \sum_{i=1}^{n}\frac{\eta^2\mu_i^2(\bar{a}_i + k_i)(B\mu_i - \ell(\bar{\boldsymbol{a}}))}{\left((\bar{a}_i + k_i)\mu_i - \ell(\bar{\boldsymbol{a}})\right)^2} \cdot \ln K,$$

*where* $\eta = \max_{i \in [n]} \alpha_i/\mu_i$, *and $c$ is a constant.*

*Proof.* We build on the techniques used in the proof of Theorem 6.1. Recall the expression of $\eta$:

$$\eta = \max_{i \in [n]} \frac{\alpha_i}{\mu_i} .$$

Define the quantities $C_{i,k}$ by

$$C_{i,k} = 2\sqrt{\frac{\ln(2k^2)}{K_{i,k}}} + 2\frac{\ln(2k^2)}{K_{i,k}} .$$

Recall that the lower confidence bounds used here are defined as

$$\hat{s}_{i,k} = \hat{\mu}_{i,k} \left(1 - \eta C_{i,k}\right)_+ \ .$$

We additionally define the following quantities

$$\hat{u}_{i,k} := \hat{\mu}_{i,k} \left(1 + \frac{4}{3}\eta C_{i,k}\right) \ .$$

We introduce the events $\mathcal{E}_{i,k}$ for $i \in [n]$ and $k \in [K]$ defined by

$$\mathcal{E}_{i,k} := \{\mu_i < \hat{s}_{i,k}\} \quad \text{or} \quad \left\{\eta C_{i,k} \leq \frac{1}{4} \quad \text{and} \quad \mu_i > \hat{u}_{i,k}\right\} \ .$$

Let

$$\mathcal{E}_k = \cup_{i \in [n]}\mathcal{E}_{i,k} \ .$$

We have, using Lemma E.3, for each $k \in [K]$ and $i \in [n]$: $\mathbb{P}\left(\mathcal{E}_{i,k}\right) \leq \frac{1}{k^2}$, which gives using a union bound $\mathbb{P}(\mathcal{E}_k) \leq \frac{n}{k^2}$.

Following similar steps as in the proof of Theorem 6.1, we call a "bad round", a round $k$ where we have $\ell(\boldsymbol{a}_k) > \ell(\bar{\boldsymbol{a}})$. Let us upper bound the number of bad rounds.

Observe that in a bad round there is necessarily an arm $i \in [K]$ such that $a_{i,k}\mu_i > \ell(\bar{\boldsymbol{a}})$. For each $i \in [n]$, let $N_i(k)$ denote the number of rounds $q \in \{1, \ldots, k\}$ where $a_{i,q}\mu_i > \ell(\bar{\boldsymbol{a}})$ and $i \in \arg\max_{j \in [n]}\{a_{j,q}\mu_j\}$ (this corresponds to a bad round triggered by worker $q$):

$$N_i(k) := |\{q \in \{1, \ldots, k\} : a_{i,q}\mu_i > \ell(\bar{\boldsymbol{a}}) \text{ and } a_{i,q}\mu_i = \ell(\boldsymbol{a}_q)\}| \ .$$

We show that in the case of $\ell(\boldsymbol{a}_k) > \ell(\bar{\boldsymbol{a}})$, the following event will hold: there exists $i \in [n]$ such that

$$E_{i,k} := \mathcal{E}_k \text{ or } \left\{N_i(k-1) \leq \frac{185\eta^2\mu_i^2(\bar{a}_i + k_i)\ln(2K^2)}{\left((\bar{a}_i + k_i)\mu_i - \ell(\bar{\boldsymbol{a}})\right)^2}\right\} \ .$$

To prove this, suppose for a contradiction argument that we have for some $i \in [n]$: $\neg E_{i,k}$ and that $k$ is a bad round triggered by arm $i$ (i.e., $\ell(\boldsymbol{a}_k) > \ell(\bar{\boldsymbol{a}})$ and $i \in \arg\max_{j \in [n]} a_{j,k}\mu_j$).

This gives in particular

$$N_i(k-1) > \frac{185\eta^2\mu_i^2(\bar{a}_i + k_i)\ln(2K^2)}{\left((\bar{a}_i + k_i)\mu_i - \ell(\bar{\boldsymbol{a}})\right)^2} \ . \tag{22}$$

Observe that in each round where $N_i(\cdot)$ is incremented, the number of samples received from the distribution $\nu_i$ increases by at least $\bar{a}_i + k_i$. Therefore, we have (22) implies

$$K_{i,k} > \frac{185\eta^2\mu_i^2(\bar{a}_i + k_i)^2\ln(2K^2)}{\left((\bar{a}_i + k_i)\mu_i - \ell(\bar{\boldsymbol{a}})\right)^2} = \frac{185\eta^2\mu_i^2\ln(2K^2)}{\left(\mu_i - \frac{\ell(\bar{\boldsymbol{a}})}{\bar{a}_i + k_i}\right)^2} \ .$$

Therefore, we have, using the expression of $C_{i,k}$ and the bound above

$$C_{i,k} = 2\sqrt{\frac{\ln(2k^2)}{K_{i,k}}} + 2\frac{\ln(2k^2)}{K_{i,k}}$$
$$\leq \frac{3}{19\eta\mu_i}\left(\mu_i - \frac{\ell(\bar{\boldsymbol{a}})}{\bar{a}_i + k_i}\right) \ . \tag{23}$$

The last bound implies in particular that $\eta C_{i,k} \leq \frac{3}{19}$, hence $(1 - \eta C_{i,k})_+ = 1 - \eta C_{i,k}$.

We have

$$
\begin{aligned}
\hat{\mu}_{i,k} - \hat{s}_{i,k} &= \hat{\mu}_{i,k} \left( 1 - (1 - \eta C_{i,k})_+ \right) \\
&\leq \eta C_{i,k} \hat{\mu}_{i,k} \\
&\leq \eta C_{i,k} \frac{\mu_i}{1 - \eta C_{i,k}} \\
&\leq \frac{1}{5} \left( \mu_i - \frac{\ell(\bar{\boldsymbol{a}})}{\bar{a}_i + k_i} \right) \ .
\end{aligned}
\tag{24}
$$

where we used the event $\neg \mathcal{E}_{i,k}$ in the penultimate inequality (in particular $\hat{s}_{i,k} = \hat{\mu}_{i,k}(1 - \eta C_{i,k})_+ \leq \mu_i$), and the bound (23) in the last inequality.

Given the hypothesis that $\ell(\boldsymbol{a}_k) > \ell(\bar{\boldsymbol{a}})$ and, $\ell(\boldsymbol{a}_k) = a_{i,k}\mu_i$, we necessarily have $a_{i,k} \geq \bar{a}_i + k_i$. Therefore, bound (24)

$$
5\hat{\mu}_{i,k} - 5\,\hat{s}_{i,k} < \mu_i - \frac{\ell(\bar{\boldsymbol{a}})}{a_{i,k}} \ .
$$

Observe that in each round $\|\boldsymbol{a}_k\|_0 = B$, therefore if we have $a_{i,k} \geq \bar{a}_i + k_i > \bar{a}_i$ for some $i$, we necessarily have that there exists $j \in [n] \setminus \{i\}$ such that $a_{j,k} \leq \bar{a}_j - 1$. Therefore, using the fact that $\ell(\bar{\boldsymbol{a}}) \geq \bar{a}_j \mu_j$, we obtain

$$
a_{i,k}(\mu_i + 5\hat{s}_{i,k} - 5\hat{\mu}_{i,k}) > \ell(\bar{\boldsymbol{a}}) \geq \bar{a}_j \mu_j \ .
\tag{25}
$$

Since both $\neg \mathcal{E}_{i,k}$ and $\neg \mathcal{E}_{j,k}$ hold (because $\neg E_{i,k}$ implies $\neg \mathcal{E}_k$), we have that

$$
\begin{aligned}
\mu_i + 5\hat{s}_{i,k} - 5\hat{\mu}_{i,k} &= \hat{s}_{i,k} + \mu_i - \hat{\mu}_{i,k} + 4\,(\hat{s}_{i,k} - \hat{\mu}_{i,k}) \\
&= \hat{s}_{i,k} + \mu_i - \hat{\mu}_{i,k} + 4\hat{\mu}_{i,k}\,((1 - \eta C_{i,k})_+ - 1) \\
&\leq \hat{s}_{i,k} + \mu_i - \hat{\mu}_{i,k} - 4\hat{\mu}_{i,k}\eta C_{i,k} \\
&\leq \hat{s}_{i,k} + \mu_i - \hat{u}_{i,k},
\end{aligned}
$$

To conclude, observe that given $\eta C_{i,k} \leq \frac{3}{19}$, event $\neg \mathcal{E}_{i,k}$ implies that $\mu_i \leq \hat{u}_{i,k}$, therefore

$$
\mu_i + 5\hat{s}_{i,k} - 5\hat{\mu}_{i,k} \leq \hat{s}_{i,k} \ .
$$

Since $\neg \mathcal{E}_{j,k}$ holds, we also have

$$
\mu_j \geq \hat{s}_{j,k} \ .
$$

Using the two last bounds in (25), we have

$$
a_{i,k}\hat{s}_{i,k} > \bar{a}_j \hat{s}_{j,k} \geq (a_{j,k} + 1)\hat{s}_{j,k} \ ,
$$

where we used the definition of $j$, as an arm satisfying $\bar{a}_j \geq 1 + a_{j,k}$, in the second inequality. This contradicts the statement of Lemma D.1, which concludes the contradiction argument. Therefore, the event that $k$ is a bad round implies that $E_{i,k}$ holds for at least one $i \in [n]$. We say that a bad round was triggered by arm $i$, a round where $N_i(\cdot)$ was incremented. Observe that if $k \in [K]$ is not a bad round then $\mathbb{E}\left[\ell(\boldsymbol{a}_k)\right] - \ell(\bar{\boldsymbol{a}}) = 0$, otherwise if $k$ is a bad round triggered by $i \in [n]$ then $\mathbb{E}\left[\ell(\boldsymbol{a}_k)\right] - \ell(\bar{\boldsymbol{a}}) \leq B\mu_i - \ell(\bar{\boldsymbol{a}})$. To ease notation we introduce for $i \in [n]$

$$
H_i := \frac{185\eta^2\mu_i^2(\bar{a}_i + k_i)\ln(2K^2)}{((\bar{a}_i + k_i)\mu_i - \ell(\bar{\boldsymbol{a}}))^2} \ .
$$

The expected regret satisfies

$$
\begin{aligned}
\mathcal{R}_K &= \sum_{i=1}^{K} \mathbb{E}\left[\ell(\boldsymbol{a}_k) - \ell(\bar{\boldsymbol{a}})\right] \\
&\leq \sum_{i=1}^{n} (B\mu_i - \ell(\bar{\boldsymbol{a}}))\mathbb{E}[N_i(K)] \\
&= \sum_{i=1}^{n} \sum_{k=1}^{K} (B\mu_i - \ell(\bar{\boldsymbol{a}}))\mathbb{E}\left[\mathbb{1}(k \text{ is a bad round triggered by } i)\right] \\
&\leq \max_{i\in[n]}\{(B\mu_i - \ell(\bar{\boldsymbol{a}}))\} \cdot \sum_{t=1}^{K} \mathbb{P}(\mathcal{E}_k) + \sum_{i=1}^{n}(B\mu_i - \ell(\bar{\boldsymbol{a}})) \sum_{k=1}^{K} \mathbb{E}\left[\mathbb{1}(k \text{ is a bad round triggered by } i) \mid \neg\mathcal{E}_k\right] \\
&\leq \max_{i\in[n]}\{(B\mu_i - \ell(\bar{\boldsymbol{a}}))\} \cdot \sum_{t=1}^{K} \mathbb{P}(\mathcal{E}_k) + \sum_{i=1}^{n}(B\mu_i - \ell(\bar{\boldsymbol{a}})) \sum_{k=1}^{K} \mathbb{E}\left[\mathbb{1}(N_i(k) = 1 + N_i(k-1) \text{ and } N_i \leq H_i) \mid \neg\mathcal{E}_k\right] \\
&\leq \max_{i\in[n]}\{(B\mu_i - \ell(\bar{\boldsymbol{a}}))\} \cdot \sum_{k=1}^{K} \mathbb{P}(\mathcal{E}_k) + \sum_{i=1}^{n}(B\mu_i - \ell(\bar{\boldsymbol{a}}))H_i \\
&\leq 2n \max_{i\in[n]}\{(B\mu_i - \ell(\bar{\boldsymbol{a}}))\} + \sum_{i=1}^{n} \frac{185\eta^2\mu_i^2(\bar{a}_i + k_i)(B\mu_i - \ell(\bar{\boldsymbol{a}}))\ln(2K^2)}{\left((\bar{a}_i + k_i)\mu_i - \ell(\bar{\boldsymbol{a}})\right)^2} \; . \qquad \square
\end{aligned}
$$

### D.3. Proof of Theorem 4.2

Let us first restate the theorem.

**Theorem 4.2.** *Suppose Assumption 3.1 holds and let $\eta := \max_{i\in[n]} \alpha_i/\mu_i$, where $\alpha_i = \|X_i - \mu_i\|_{\psi_1}$. Then, the total expected computation time after $K$ rounds, using the allocation prescribed by* ATA *with inputs $(B, \alpha)$ satisfies*

$$
\mathcal{C}_K \leq (1 + 4\eta\ln(B))\,\mathcal{C}_K^* + \mathcal{O}(\ln K) \; .
$$

*Proof.* Let $\mathbb{E}_k$ be the expectation with respect to the variables observed up to and including $k$ and $\mathcal{F}_k$ the corresponding filtration. Using the tower rule, we have

$$
\sum_{k=1}^{K} \mathbb{E}\left[C(\boldsymbol{a}_k)\right] = \mathbb{E}\left[\sum_{k=1}^{K} \mathbb{E}_{k-1}[C(\boldsymbol{a}_k)]\right] \; .
$$

Consider round $k \in [K]$, let us upper bound $\mathbb{E}_{k-1}[C(\boldsymbol{a}_t)]$ using $\mathbb{E}_{k-1}[\ell(\boldsymbol{a}_k)]$. Recall that $\boldsymbol{a}_k \in \mathcal{F}_{k-1}$, let $Y_i = \sum_{u=1}^{a_{i,k}} X_{i,k}^{(u)}$, since $Y_i$ is the sum of $a_{i,k}$ i.i.d samples we have that $\mathbb{E}_{k-1}[Y_i] = a_{i,k}\mu_i$ and $\|Y_i - a_{i,k}\mu_i\|_{\psi_1} \leq a_{i,k}\|X_i - \mu_i\|_{\psi_1}$. Thus, using Lemma E.4, we get

$$
\begin{aligned}
\mathbb{E}_{k-1}\left[C(\boldsymbol{a}_k)\right] &= \mathbb{E}_{k-1}\left[\max_{i\in\text{supp}(\boldsymbol{a}_k)} \left\{\sum_{u=1}^{a_{i,k}} X_{i,k}^{(u)}\right\}\right] \\
&\leq \max_{i\in\text{supp}(\boldsymbol{a}_k)}\{a_{i,k}\mu_i\} + 4 \max_{i\in\text{supp}(\boldsymbol{a}_k)}\{a_{i,k}\alpha_i\} \cdot \ln(B) \\
&\leq \max_{i\in\text{supp}(\boldsymbol{a}_k)}\{a_{i,k}\mu_i\} + 4 \max_{i\in\text{supp}(\boldsymbol{a}_k)}\{a_{i,k}\,\eta\mu_i\} \cdot \ln(B) \\
&= (1 + 4\eta\ln(B)) \max\{a_{i,k}\mu_i\} \; .
\end{aligned}
$$

Moreover, using Jensen's inequality, we have

$$
\max_{i\in[n]}\{a_i^*\mu_i\} \leq \mathbb{E}\left[\max_{i\in[n]} \left\{\sum_{u=1}^{a_{k,i}} X_{i,k}^{(u)}\right\}\right] = \mathbb{E}[C(\boldsymbol{a}^*)] \; .
$$

Using the last two bounds with the result of Theorem 6.1, we get the result.

$$\square$$

# E. Technical Lemmas

The lemma below gives a concentration bound on sub-exponential variables. Note that this result can be inferred from Proposition 7 in Maurer & Pontil (2021), although applying the last result directly requires assuming the variables are positive, this is not needed in their proof in the one dimensional case. For completeness, we present the full proof below.

**Lemma E.1.** *Let $Y_1, \ldots, Y_n$ be i.i.d random variables with $\mathbb{E}[Y_1] = 0$ and $\alpha = \|Y_1\|_{\psi_1} < +\infty$. Then for any $\delta \in (0, 1)$, we have with probability at least $1 - \delta$*

$$\left| \frac{1}{n} \sum_{i=1}^{n} Y_i \right| \leq 2\alpha \left( \sqrt{\frac{\ln(2/\delta)}{n}} + \frac{\ln(2/\delta)}{2n} \right) .$$

*Proof.* Let $v := 2n\alpha^2$. We have using Lemma E.5 that

$$\sum_{i=1}^{n} \mathbb{E}[Y_i^2] \leq 2n\alpha^2 \quad \text{and} \quad \sum_{i=1}^{n} \mathbb{E}[(Y_i)_+^q] \leq \frac{q!}{2} v\alpha^{q-2} .$$

Therefore, using Bernstein concentration inequality (Proposition E.2) we obtain that

$$\mathbb{P}\left( \left| \sum_{i=1}^{n} Y_i \right| \geq 2\alpha\sqrt{nt} + \alpha t \right) \leq 2\exp(-t) .$$

Choosing $t = \ln(2/\delta)$, we obtain the result. $\square$

**Proposition E.2** (Theorem 2.10 in Boucheron et al. (2005)). *Let $X_1, \ldots, X_n$ be independent real-valued random variables. Assume there exist positive numbers $v$ and $c$ such that*

$$\sum_{i=1}^{n} \mathbb{E}[X_i^2] \leq v \quad \text{and} \quad \sum_{i=1}^{n} \mathbb{E}[(X_i)_+^q] \leq \frac{q!}{2} v\, c^{q-2}, \qquad \text{for all integers } q \geq 3,$$

*where $x_+ := \max\{x, 0\}$. Define the centered sum*

$$S := \sum_{i=1}^{n} (X_i - \mathbb{E}X_i).$$

*Then, for every $t > 0$,*

$$\mathbb{P}\left( S \geq \sqrt{2vt} + ct \right) \leq e^{-t}.$$

**Lemma E.3.** *Let $X_1, \ldots, X_n$ be i.i.d positive random variables with mean $\mu$ and $\|X_1\|_{\psi_1} < +\infty$. Denote $\alpha := \|X_1 - \mu\|_{\psi_1}$. Denote $\hat{X}_n = \frac{1}{n} \sum_{i=1}^{n} X_i$ and let $\delta \in (0, 1)$. Define $\eta, C_{\cdot,\cdot}$ by:*

$$\eta := \frac{\alpha}{\mu} \quad \text{and} \quad C_{n,\delta} := 2\sqrt{\frac{\ln(2/\delta)}{n}} + 2\frac{\ln(2/\delta)}{n} .$$

*Then with probability at least $1 - \delta$ we have*

$$\mu \geq \hat{X}_n (1 - \eta \cdot C_{n,\delta})_+ ,$$

*where we use the notation $(a)_+ = \max\{0, a\}$. Moreover, if $\eta\, C_{n,\delta} \leq \frac{1}{4}$, then with probability least $1 - \delta$*

$$\hat{X}_n (1 - \eta \cdot C_{n,\delta})_+ \leq \mu \leq \hat{X}_n \left( 1 + \frac{4}{3}\eta \cdot C_{n,\delta} \right) .$$

*Proof.* Fix $n, \delta$. We work on the event

$$\mathcal{E}_{n,\delta} = \left\{ \left| \hat{X}_n - \mu \right| \leq \alpha \cdot C_{n,\delta} \right\}$$

that holds with probability at least $1 - \delta$ if we apply Proposition E.1 to $X_i - \mu$.

**Proof of** $\mu \geq \hat{X}_n (1 - \eta \cdot C_{n,\delta})_+$**:**

If $\eta C_{n,\delta} \geq 1$, we have $(1 - \eta \cdot C_{n,\delta})_+ = 0$ and the result follows from the fact that $X$ is non-negative which gives $\mu \geq 0$.

Suppose now that $\eta C_{n,\delta} < 1$. Recall that event $\mathcal{E}_{n,\delta}$ implies that

$$\hat{X}_n \leq \mu(1 + \eta C_{n,\delta}) .$$

Therefore, we have

$$\frac{\hat{X}_n}{1 + \eta C_{n,\delta}} \leq \mu .$$

Using $1 - \eta C_{n,\delta} \leq \frac{1}{1 + \eta C_{n,\delta}}$ with the bound above, we obtain

$$\hat{X}_n (1 - \eta C_{n,\delta})_+ = \hat{X}_n (1 - \eta C_{n,\delta}) \leq \frac{\hat{X}_n}{1 + \eta \cdot C_{n,\delta}} \leq \mu .$$

**Proof of** $\mu \leq \left(1 + \frac{4}{3}\eta \cdot C_{n,\delta}\right)$**.** Recall that event $\mathcal{E}_{n,\delta}$ gives

$$\hat{X}_n \geq \mu - \alpha C_{n,\delta} = \mu(1 - \eta C_{n,\delta}) .$$

Suppose that $\eta C_{n,\delta} \leq \frac{1}{4}$. We therefore have

$$\mu \leq \frac{\hat{X}_n}{1 - \eta C_{n,\delta}} .$$

Next, we use the fact that for any $x \in [0, 1/4]$, we have

$$\frac{1}{1 - x} \leq 1 + \frac{4}{3}x ,$$

which gives

$$\mu \leq \hat{X}_n \left(1 + \frac{4}{3}\eta \cdot C_{n,\delta}\right) ,$$

when $\eta C_{n,\delta} \leq \frac{1}{4}$. $\qquad\square$

**Lemma E.4.** *Let $X_1, \ldots, X_n$ be a sequence of independent random variables with finite Orlicz norm $\|X_i\|_{\psi_1} < +\infty$ and let $\mathbb{E}[X_i] = \mu_i$. Then we have*

$$\mathbb{E}[\max_{i \in [n]} X_i] \leq \max_{i \in [n]} \mu_i + 4\alpha \ln(n) ,$$

*where $\alpha = \max_{i \in [n]} \|X_i - \mu_i\|_{\psi_1}$.*

*Proof.* If $n = 1$ the bound is straightforward, suppose that $n \geq 2$. Let $Y_i := X_i - \mu_i$, then $\alpha = \max_{i \in [n]} \|Y_i\|_{\psi_1}$. Let $\lambda \in (0, 1/\alpha)$, we have

$$\max_{i \in [n]} Y_i = \frac{1}{\lambda} \ln \left( \exp \left( \lambda \max_{i \in [n]} Y_i \right) \right)$$

$$\leq \frac{1}{\lambda} \ln \left( \sum_{i=1}^{n} \exp (\lambda Y_i) \right) .$$

Taking the expectation and using Lemma E.5, we have

$$\mathbb{E}\left[\max_{i \in [n]} Y_i\right] \leq \frac{1}{\lambda} \ln \left( \sum_{i=1}^{n} \mathbb{E}\left[\exp(\lambda Y_i)\right] \right)$$

$$\leq \frac{1}{\lambda} \ln \left( \frac{n}{1 - \lambda \alpha} \right) .$$

We choose $\lambda = \frac{1-1/n}{\alpha}$, which gives

$$
\begin{aligned}
\mathbb{E}\left[\max_{i\in[n]} Y_i\right] &\leq \frac{\alpha}{1-\frac{1}{n}} \ln(n^2) \\
&= 2\alpha \frac{n}{n-1} \ln(n) \\
&\leq 4\alpha \ln(n) .
\end{aligned}
\tag{26}
$$

Let $i^* \in \arg\max_{i\in[n]} X_i$, we have

$$
\begin{aligned}
\max_{i\in[n]} X_i - \max_{i\in[n]} \mu_i &= X_{i^*} - \max_{i\in[n]} \mu_i \\
&\leq X_{i^*} - \mu_{i^*} \leq \max_{i\in[n]}\{X_i - \mu_i\} = \max_{i\in[n]} Y_i .
\end{aligned}
$$

Combining the last bound with (26) we obtain

$$
\mathbb{E}\left[\max_{i\in[n]} X_i\right] \leq \max_{i\in[n]} \mu_i + 4\alpha \ln(n) .
$$

$\square$

Lemma below is based on a standard argument we give here for completeness.

**Lemma E.5.** *Let $Y$ be a variable such that $\alpha = \|Y\|_{\psi_1} < +\infty$. Then we have for any $\lambda \in \left(-\frac{1}{\alpha}, \frac{1}{\alpha}\right)$*

$$
\mathbb{E}\left[\exp\left(\lambda Y\right)\right] \leq \frac{1}{1 - |\lambda|\,\alpha} .
$$

*Moreover, we have for any $q \geq 3$*

$$
\mathbb{E}\left[Y^2\right] \leq 2\alpha^2 \quad \text{and} \quad \mathbb{E}\left[(Y)_+^q\right] \leq \frac{q!}{2} \cdot (2\alpha^2) \cdot \alpha^{q-2} .
$$

*Proof.* Let $Z = |Y|/\alpha$. First observe that we have

$$
\sum_{k\geq 0} \frac{\mathbb{E}\left[Z^k\right]}{k!} = \mathbb{E}\left[\exp(Z)\right] = \mathbb{E}\left[\exp(|Y|/\alpha)\right] \leq 2 ,
$$

so,

$$
\sum_{k\geq 1} \frac{\mathbb{E}\left[Z^k\right]}{k!} \leq 1 .
$$

This implies $\mathbb{E}\left[|Y|^k\right] \leq k!\alpha^k$ for all $k \geq 1$. Using this bound, we estimate the moment generating function

$$
\begin{aligned}
\mathbb{E}\left[\exp(\lambda Y)\right] &\leq \mathbb{E}\left[\exp\left(|\lambda|\,|Y|\right)\right] \\
&= \sum_{k\geq 0} \frac{|\lambda|^k \mathbb{E}\left[|Y|^k\right]}{k!} \\
&\leq 1 + \sum_{k\geq 1} |\lambda|^k \alpha^k \\
&= \frac{1}{1 - |\lambda|\,\alpha} .
\end{aligned}
$$

The remaining bounds follow from $\mathbb{E}\left[|Y|^k\right] \leq k!\alpha^k$. $\square$

