# OpenReview forum: "ATA: Adaptive Task Allocation for Efficient Resource Management in Distributed Machine Learning"
_ICML.cc/2025/Conference — ICML 2025 poster_

### Official Review · Reviewer_bw5d · 2025-02-27

**Overall Recommendation:** 3

**Summary:**

In distributed machine learning, traditional greedy methods waste resources by assigning tasks to every worker, even when only a few results are needed. This paper tells a story of efficiency: it introduces ATA, a smart task allocator that learns the speed of each worker on the fly. Instead of needing prior knowledge, ATA treats task allocation like a multi-armed bandit problem, using lower confidence bounds to decide where to assign tasks. By balancing exploration (learning each worker’s speed) and exploitation (favoring faster workers), ATA nearly matches the performance of an ideal, all-knowing strategy while eliminating wasted work. An improved variant, ATA-Empirical, tailors this approach even further to individual workers, achieving better results in practice. Overall, the paper offers a principled and practical solution to cut waste and boost efficiency in large-scale machine learning.

**Claims And Evidence:**

The paper claims that ATA dynamically learns worker speeds and allocates tasks so as to minimize total computation time without prior knowledge, nearly matching the performance of an ideal oracle. It recasts task allocation as a combinatorial bandit problem with rigorous theoretical guarantees, ensuring logarithmic regret and near-optimal performance, and shows through experiments that both ATA and its variant, ATA-Empirical, substantially reduce wasted computation compared to greedy methods.

**Essential References Not Discussed:**

n/a

**Experimental Designs Or Analyses:**

The experiments of Figure 1 are conducted on a variety of algorithms and use a wide range of metrics to measure the effectiveness of these algorithms. While I find these results interesting, they lack real-world experiments on real-world benchmarks such as MLPerf.

**Methods And Evaluation Criteria:**

Evaluation is carried out through simulations on distributed SGD tasks, comparing the proposed methods against baselines such as Greedy Task Allocation, Uniform Task Allocation, and an oracle-based fixed strategy. Key metrics include runtime efficiency, total worker time, average iteration time, and cumulative regret.

**Other Comments Or Suggestions:**

n/a

**Other Strengths And Weaknesses:**

n/a

**Questions For Authors:**

n/a

**Relation To Broader Scientific Literature:**

The paper builds on asynchronous optimization methods like Rennala SGD and Ringmaster ASGD by addressing their resource inefficiencies through a novel formulation of task allocation as a combinatorial multi-armed bandit problem. It leverages foundational bandit theory, particularly lower confidence bounds, to balance exploration and exploitation without prior knowledge of worker speeds, while also resonating with federated learning approaches that adjust computations for heterogeneous devices. This integration of ideas from distributed optimization, bandit theory, and federated learning marks a significant advancement in tackling resource waste in large-scale machine learning.

**Theoretical Claims:**

n/a

---

> ### Author Rebuttal · Authors · 2025-03-30
>
> Thank you for the review
>
> > they lack real-world experiments on real-world benchmarks such as MLPerf.'}
>
> There might be a misunderstanding on the nature of our contribution: We propose an allocation strategy across workers that adapts to their computation times and that is *agnostic* to the specifics of the task.
> This means that the behaviour of our algorithm does not depend on the optimization algorithm used, the dataset, the loss function, etc.
> Instead, it *only* depends on the observed computation times. This means that, given a sequence of computation times for any number of workers and a training loss curve, anyone can immediately simulate how our algorithm would affect the total computation time. Indeed, this would even work for other tasks different than optimization.
> Besides, MLPerf does not contain benchmarks for distributed optimization or bandit algorithms.
>
> That said, to answer the reviewer's point, we ran additional experiments using a CNN trained with Adam on the CIFAR-100 dataset.
> The plots can be found
> [here](https://anonymous.4open.science/api/repo/real_data-3BF5/file/real_data.pdf?v=33782a9d).

---

> > ### Comment · Reviewer_bw5d · 2025-04-01
> >
> > Thanks you for the additional details and results. This satisfies my concerns and therefore I will raise my score.

---

### Official Review · Reviewer_svBT · 2025-03-09

**Overall Recommendation:** 4

**Summary:**

This paper proposes a new task allocation algorithm for efficiently managing computation resources in distributed SGD.

**Claims And Evidence:**

Yes.

**Essential References Not Discussed:**

No.

**Experimental Designs Or Analyses:**

Authors implemented simulations to evaluate the validity of the proposed algorithm. But there is no real dataset experiment.

The paper lacks explanation of parameter settings in section 7 (experiment part).

**Methods And Evaluation Criteria:**

Yes.

**Other Comments Or Suggestions:**

Could you add some simple experiments for real dataset? Please explain the parameter settings in section 7.

**Other Strengths And Weaknesses:**

Strength: The proposed algorithm has solid theoretical analysis. The authors innovatively reduce the task allocation problem to a MAB problem.

Weaknesses:
1. The paper only shows the simulation results, but there is no real dataset experiment (e.g. distributed deep learning training allocation) to show the performance of their approach.
2. Given that the computation time of worker nodes may be affected by dynamic load, task switching, etc. The distribution assumption and bound of the nodes may not be practical in real-world scenario.
3. It seems that there is no computational complexity analysis of the proposed method, and the comparison with baselines.
4. The experiment section lacks explanation of parameter settings.

**Questions For Authors:**

No more questions.

**Relation To Broader Scientific Literature:**

The paper reduces task allocation problems to an online bandit problem (MAB problem), and utilizes proxy loss to solve the computational challenge caused by the combinatorial nature of the action set.

**Theoretical Claims:**

I have read the theoretical claims in the main part. The proofs in the appendix are not checked.

---

> ### Author Rebuttal · Authors · 2025-03-30
>
> We would like to thank the reviewer for their valuable feedback.
>
> > The paper only shows the simulation results, but there is no real dataset experiment (e.g. distributed deep learning training allocation) to show the performance of their approach.
>
> There might be a misunderstanding on the nature of our contribution: We propose an allocation strategy across workers that adapts to their computation times and that is *agnostic* to the specifics of the task.
> This means that the behaviour of our algorithm does not depend on the optimization algorithm used, the dataset, the loss function, etc.
> Instead, it *only* depends on the observed computation times. This means that, given a sequence of computation times for any number of workers and a training loss curve, anyone can immediately simulate how our algorithm would affect the total computation time. Indeed, this would even work for other tasks different than optimization.
>
> That said, to answer to the reviewer's point, we ran additional experiments using a CNN trained with Adam on the CIFAR-100 dataset.
> The plots can be found
> [here](https://anonymous.4open.science/api/repo/real_data-3BF5/file/real_data.pdf?v=33782a9d).
>
> > Given that the computation time of worker nodes may be affected by dynamic load, task switching, etc. The distribution assumption and bound of the nodes may not be practical in real-world scenario.
>
> We acknowledge the reviewer's point that incorporating dynamic load and task switching would align the model more closely with practical scenarios. However, to the best of our knowledge, this is the first work on adaptive online allocation under a limited budget that provides strong theoretical guarantees on the optimality of the allocation strategy. It seems a bit unfair to require the solution of all the possible problems in a single first paper.
> The assumptions made regarding the distribution of computing times are standard and commonly adopted in the literature for modeling such quantities. We leave extending the framework to accommodate more general and irregular distributions for future work.
>
> > It seems that there is no computational complexity analysis of the proposed method, and the comparison with baselines.
>
> We do discuss the computational complexity, but maybe we could stress it more. Specifically, for our main procedures, ATA and ATA-Empirical, the computational complexity per round is primarily determined by the step of computing the allocation $a_k$ (line 5 in Algorithm 1). As stated in lines 205-208, the complexity of this step is linear in $n$ when the number of workers $n$ exceeds the task budget $B$, demonstrating the efficiency of our approach. For more details please check Section C of the appendix.
>
> > The experiment section lacks explanation of parameter settings.
>
> Thank you for pointing this out. We noticed that the dimensionality and stepsize were missing. For our experiments, we chose $d=100$ and a stepsize of $\gamma=10^{-4}$. Is there anything else we may have missed? We will include all the missing information in the camera-ready version of the paper.

---

### Official Review · Reviewer_9eUV · 2025-03-13

**Overall Recommendation:** 3

**Summary:**

This paper suggests an adaptive method for task allocation, Adaptive Task Allocation (ATA). In the context of machine learning, this approach addresses federated learning, where in each round, $n$ workers collaboratively perform minibatch SGD with total batch size of $B$. In contrast to the standard approach, where each worker processes a fixed minibatch of size $B/n$, the proposed approach aims to provide a more efficient allocation of minibatch sizes across workers that minimizes the overall runtime per round without unnecessary computational overhead. The paper provides theoretical guarantees in the framework of online convex bandit, and presents empirical results to validate the effectiveness of the proposed approach.

**Claims And Evidence:**

The proofs of the theoretical claims were not checked, and the empirical results may be less convincing in realistic scenarios (see "Methods And Evaluation Criteria" and "Weaknesses").

**Essential References Not Discussed:**

N/A

**Experimental Designs Or Analyses:**

I reviewed the experimental design and analysis in the paper (issues were discussed at "Methods And Evaluation Criteria" and "Weaknesses").

**Methods And Evaluation Criteria:**

The authors evaluated their approach using regret analysis, which is standard for bandit applications. They also provided an empirical evaluation in a simple convex optimization setting; however, given that this is for a machine learning publication, a thorough evaluation (and adaptation) using standard benchmark datasets and more complex objective functions would be more appropriate.

**Other Comments Or Suggestions:**

* The case where $s_{i,k} = 0$ for some worker $i$ is only discussed in the appendix on page 16, lines 841–842, instead of at the main paper.

* To avoid being misleading, the authors should explicitly include the dependence of Compute LCBs on $\alpha$ or $\eta$ in the proposed algorithms (such as in Algorithm 1).

**Other Strengths And Weaknesses:**

*Strengths*:
1. I found it interesting to approach this problem from the perspective of a bandit problem, which, on one hand, exploits fast workers and, on the other hand, allows for exploring changes in the performance of weaker workers, whose running time can vary over time. This can be realistic in some practical scenarios. The proposed approach might be a good first direction, but modifications in the confidence assessment may be necessary for realistic and more complex problems (see weaknesses).

2. This idea can be relevant today in the context of energy efficiency, as GPUs are used at scale and consume significant amounts of power.

*Weaknesses*:
1. In the case where $s_{i,k} = 0$ for some worker $i$,  the authors used a suboptimal uniform allocation (see "Other Comments Or Suggestions") instead of an impractical optimal allocation (e.g., assigning $B$ allocations to some worker $i$ with $ s_{i,k}= 0$ and $0$ to the others). How does this choice affect the theoretical results?

2. ATA-Empirical relies on the parameter $\eta$, which is unknown in practice and needs to be evaluated. This can be critical, as large $\eta$ may cause $s_{i,k}$ to become zero for some worker $i\in[n]$, which leads to a suboptimal uniform allocation.

3. The confidence assessment depends on the number of iterations $k$, $K$. While using these parameters might work for small and simple problems, as demonstrated in the experiments, it may be less practical in more complex scenarios. In such cases, the confidence assessment may not align with the actual changes in workers' running times, and make it less meaningful.

**Questions For Authors:**

see weaknesses.

**Relation To Broader Scientific Literature:**

N/A

**Theoretical Claims:**

I didn't check the correctness of the proofs; the complete proofs are in the appendix and weren't reviewed.

---

> ### Author Rebuttal · Authors · 2025-03-30
>
> Thank you for the review
>
> > The paper provides theoretical guarantees in the framework of online convex bandit
>
> Since the action set is discrete, the proposed reduction does not fall under the framework of online convex bandits but rather corresponds to a combinatorial bandit problem.
>
> > however, given that this is for a machine learning publication, a thorough evaluation (and adaptation) using standard benchmark datasets and more complex objective functions would be more appropriate.
>
> Please take a look at our response to reviewer bw5d. We also provide additional experiments
> [here](https://anonymous.4open.science/api/repo/real_data-3BF5/file/real_data.pdf?v=33782a9d).
>
> > Weakness 1
>
> This choice does not affect the theoretical results.
> More in detail, in the initial phase of the procedure, due to the definition $s_{i,k} = (\hat{\mu}\_{i,k} - \text{conf}(i,k))\_{+}$, all workers satisfy  $s\_ {i, k} = 0$. After this warm-up stage, the scores $s_{i,k}$ become strictly positive. Recall that at round $k$, $\hat{\mu}\_{i,k}$ denotes the empirical mean of the computation times for worker $i$, and $\text{conf}(i,k)$ is a confidence term of order $\alpha^2 \sqrt{\frac{\ln(2k^2)}{K{i,k}}}$, where $K_{i,k}$ is the number of times worker $i$ has been queried up to round $k$. This implies that the number of rounds for which $s_{i,k} = 0$ is at most $\alpha^2 \ln(k) / \mu_i^2$.
>
> Furthermore, in the theoretical guarantees--specifically in the upper bound of Theorem 6.1 (see also Equation (11))--the number of rounds in which worker $i$ incurs a loss due to overallocation is of order $\alpha^2 \ln(k)/\left(\mu_i - \frac{\ell(\bar{a}, \mu)}{\bar{a}i + k_i}\right)^2$. This shows that the losses incurred during the warm-up stage, where $s_{i,k} = 0$, are dominated by those arising when $s_{i,k} > 0$ for all $i$. A more detailed analysis is provided in the proof of Theorem 6.1 (Section D.1).
>
> In practice, during the warm-up phase when many $s_{i,k} = 0$, the allocation $a_k$ (line 5 of Algorithm 1) is not uniquely defined. As noted by the reviewer, we resolve this by uniformly distributing the workload among the workers with $s_{i,k} = 0$. However, any heuristic that ensures $a_k \in \arg\min_a \ell(a, s_k)$ could be used without any change to the theoretical guarantees. We chose the uniform strategy for its simplicity, and as our experiments show, it applies only during a small fraction of the early rounds. We will add this clarification as a remark in the main body of the paper.
>
>
> > Weakness 2
>
> This is a limitation of virtually any bandit algorithm. In fact, in the case of bounded distributions, many existing procedures assume knowledge of the range of the losses. Moreover, we do not believe it is possible to design a completely parameter-free procedure for the problem we considered because proving any concentration result on the empirical mean of a random variable without any assumption on its variance is known to be impossible.
>
> That said, we would like to draw the reviewer's attention to the fact that ATA-Empirical requires only $\eta$ as input, instead of $\alpha$. That is, each of our algorithms requires only a single distribution-dependent parameter. For example, in cases where the learner knows that the computation times approximately follow an exponential distribution--which is a common modeling assumption (see lines 152–160)--it is advantageous to use ATA-Empirical, since $\eta$ is necessarily of constant order (i.e., $O(1)$).
>
> Regarding the warm-up phase where $s_{i,k} = 0$: as with ATA, in ATA-Empirical, a larger value of $\eta$ results in a longer initial phase during which $s_{i,k} = 0$. However, the previous observation still holds: The loss incurred during this stage is asymptotically dominated by the regret bound guaranteed by the algorithm.
>
> > Weakness 3
>
> We are not entirely certain we understand the reviewer’s point. The confidence bounds used for the computation times *increase* with the number of iterations (ensuring exploration) and *naturally shrink* with the number of observed samples from the corresponding distribution (this quantity is denoted $K_{i,k}$ for worker $i$ up to round $k$). These bounds are derived using concentration inequalities for sub-exponential distributions (see Proposition E.1).
>
> Also, our theoretical guarantees are established under the assumption of stationary distributions, hence, the expected value of the workers' running times cannot change. We believe that, given this is--to the best of our knowledge--the first contribution addressing adaptive allocation with partial feedback and theoretical guarantees, it is both natural and appropriate to begin with the stationary setting. This assumption is also standard in the majority of bandit literature. We leave extending the analysis to the non-stationary case, where workers' distributions may vary over time, as an important direction for future work.

---

### Official Review · Reviewer_xCoQ · 2025-03-17

**Overall Recommendation:** 3

**Summary:**

This paper introduces ATA (Adaptive Task Allocation), a method that dynamically adapts to heterogeneous and random worker computation times without prior knowledge of their distributions. Theoretical analysis shows that ATA achieves optimal task allocation, performing as well as methods that have prior knowledge of computation times. Experimental results confirm that ATA significantly reduces resource costs compared to greedy methods, which can become arbitrarily expensive as the number of workers increases.

**Claims And Evidence:**

The general task allocation problem in parallel stochastic optimization leads to resource wastefulness, and addressing it can improve efficiency across various distributed machine learning methods. Supported by many asynchronous SGD methods such as Ringmaster ASGD and Rennala SGD.

**Essential References Not Discussed:**

Most of recent works are all referenced.

**Experimental Designs Or Analyses:**

For different number of workers, they measured runtime vs suboptimality, suboptimality against total worker time, average iteration time and averaged cumulative regret.

**Methods And Evaluation Criteria:**

**Methods** This work formalizes the task allocation problem as a combinatorial online learning problem with partial feedback and non-linear losses. The authors propose ATA, a lower-confidence bound-based algorithm that is agnostic to workers' computation times and achieves near-optimal computation efficiency. They also introduce ATA-Empirical, a data-driven variant that improves empirical performance, and validate their approach through numerical simulations.
**Evaluation** Authors test the algorithm by simulating a scenario with n workers. They collect 23 gradients from the workers and perform gradient descent step. They compare with GTA, and UTA.

**Other Comments Or Suggestions:**

Please see above

**Other Strengths And Weaknesses:**

(+) solving this problem in online bandit problem is interesting.
(+) strong theoretical support on the proposed algorithm
(-) missing ablation study (e.g., impact of data-dependent concentration inequality), realistic simulation (when workers have heterogeneous and time-vayring computation speeds)

**Questions For Authors:**

Is there any case that information of previous runs degrade performance or slow down the convergence?
Could you elaborate more on the impact of using prior information?

**Relation To Broader Scientific Literature:**

It mainly focus on asynchronous training field.

**Theoretical Claims:**

Proposed algorithms and theorems are backed by theorietical proof.

---

> ### Author Rebuttal · Authors · 2025-03-30
>
> We would like to thank the reviewer for their valuable feedback.
>
> > missing ablation study (e.g., impact of data-dependent concentration inequality)
>
> We are not sure what the reviewer means here: We did study the impact of the data-dependent concentration inequality. Indeed, we test both ATA (the algorithm without the adaptive concentration) and ATA-Empirical (the one with the adaptive one).
>
> > realistic simulation (when workers have heterogeneous and time-vayring computation speeds)
>
> We did additional experiments with heterogeneous time distributions for the workers, using five types: Exponential, Uniform, Half-Normal, Lognormal, and Gamma.
> The plots are available
> [here](https://anonymous.4open.science/api/repo/hetero_plots-37B3/file/hetero_plots.pdf?v=e4d1e678).
> The results show that the algorithms are robust across different distributions.
>
> Also, the computation speeds are time-varying because they are stochastic. It is true that we did not consider the setting of distribution changing over time, however, we believe that, given this is -to the best of our knowledge- the first contribution addressing adaptive allocation with partial feedback and theoretical guarantees, it is both natural and appropriate to begin with the stationary setting.
>
> > Is there any case that information of previous runs degrade performance or slow down the convergence? Could you elaborate more on the impact of using prior information?
>
> On page 8, we discussed the possibility of using prior estimates of computation times from previous runs.
> We add more details here on this possibility:
> If the prior estimates are completely off, the algorithm could make wrong choices in the first rounds, however sooner or later the exploration (that is always active) will cause the algorithm to gather enough data to converge to the correct estimate of the means. Hence, the asymptotic guarantee will be the same and prior knowledge almost always accelerates convergence.
> To validate this, we conducted additional experiments, please check the plots
> [here](https://anonymous.4open.science/api/repo/prior_knowledge-2E5C/file/prior_knowledge.pdf?v=b620dfe3).

---

### Decision · Program_Chairs · 2025-05-01

**Decision:**

Accept (poster)

**Comment:**

This paper proposes an adaptive method for task allocation, which has an application in collaboratively performing mini-batch SGD with a fixed total batch size. More precisely, unlike standard approaches where each worker processes an equal-sized mini-batch, the proposed adaptive task allocation method dynamically allocates mini-batch sizes across workers to minimize the overall runtime per round, while avoiding unnecessary computational overhead. The underlying technique is a bandit algorithm. The topic is highly relevant to the optimization and machine learning community. All reviewers unanimously recommend acceptance of this work.

The authors are expected to include implementation details for their real-world dataset experiments in the rebuttal phase and incorporate the constructive feedback that they received here in the final version.